# Importin 13-dependent axon diameter growth regulates conduction speeds along myelinated CNS axons

Jenea M. Bin [1] ✉, Daumante Suminaite[1], Silvia K. Benito-Kwiecinski[1], Linde Kegel [1], Maria Rubio-Brotons[1], Jason J. Early[1], Daniel Soong [1], Matthew R. Livesey[1,2,3], Richard J. Poole [4] & David A. Lyons [1] ✉

Axon diameter influences the conduction properties of myelinated axons, both directly, and indirectly through effects on myelin. However, we have limited understanding of mechanisms controlling axon diameter growth in the central nervous system, preventing systematic dissection of how manipulating diameter affects myelination and conduction along individual axons. Here we establish zebrafish to study axon diameter. We find that importin 13b is required for axon diameter growth, but does not affect cell body size or axon length. Using neuron-specific *ipo13b* mutants, we assess how reduced axon diameter affects myelination and conduction, and find no changes to myelin thickness, precision of action potential propagation, or ability to sustain high frequency firing. However, increases in conduction speed that occur along single myelinated axons with development are tightly linked to their growth in diameter. This suggests that axon diameter growth is a major driver of increases in conduction speeds along myelinated axons over time.

Neurons of the vertebrate central nervous system (CNS) have axons with incredibly diverse diameters, ranging from -0.1 to >10 μm, which corresponds to over ten thousand-fold differences in cross-sectional area[1–3]. The majority of this spectrum of diameters is represented by axons that are myelinated, with biophysical and molecular signals associated with axon diameter being critical cues for myelination[4–9]. The diversity of myelinated axon diameter is key in shaping neuronal circuit function, as axon diameter affects conduction speed and has also been proposed to influence action potential firing rates[3]. Studies comparing myelinated axons of different diameters have found that, in general, axon diameter scales proportionally with the speed of action potential propagation, such that conduction velocities, like axon diameter, also vary by over 100-fold[10,11]. While axon diameter has a direct effect on conduction speeds, this linear relationship is also due in part to the scaling of myelin thickness with axon diameter across axons[10,12,13]. However, variation to multiple parameters along the myelinated axon (e.g.

axon diameter, myelin sheath length and thickness, dimensions of nodes of Ranvier, expression and density of ion channels) can all modulate conduction[10,14,15], and in the context of an individual axon, it remains unclear how changes to its diameter actually affect its myelin and conduction properties over time. This is particularly relevant during development, as many axons continue to grow in diameter after they become myelinated, with some axons growing well into adolescence[16,17]. More recent studies have also indicated that axon diameter may be dynamically regulated throughout life in response to neuronal activity[18–20], while others have shown that axon diameter becomes disrupted across a wide variety of neuro-developmental, neuropsychological, and neurodegenerative diseases[21–28]. Thus, it is important to understand how axon diameter is regulated and how changes to axon diameter affect structural and functional parameters of individual myelinated axons if we want to gain a full understanding of how the precise timings of signal propagation required for proper nervous system function are achieved.

[1]Centre for Discovery Brain Sciences, University of Edinburgh, Edinburgh EH16 4SB, UK. [2]Sheffield Institute for Translational Neuroscience, University of Sheffield, Sheffield S10 2HQ, UK. [3]Neuroscience Institute, University of Sheffield, Sheffield S10 2TN, UK. [4]Department of Cell and Developmental Biology, University College London, London WC1E 6BT, UK. ✉e-mail: jbin@exseed.ed.ac.uk; david.lyons@ed.ac.uk

Despite its importance, little is known about how axon diameter is established and regulated compared to other aspects of neuronal biology, and thus few experimental models exist with which to test the specific role of axon diameter on myelinated axon structure and function. The majority of mechanistic studies have focussed on cytoskeletal components such as neurofilaments, microtubules, and actin-spectrin rings that influence axon diameter[29–35], but the cell-intrinsic programmes or cell–cell interactions that promote or inhibit diameter growth in the CNS remain largely unexplored. General growth pathways that regulate cell size can result in changes to axon diameter[5,36–38], but are at least partially separable from the specific regulation of axon diameter growth over time. Indeed, not only have disconnects between the growth of the cell body and axon diameter been observed, but individual axons can vary in diameter along their length, or across their branches[39–41]. Furthermore, technical challenges in visualising, measuring, and following changes to axon diameter along individual axons over time in in vivo model systems have prevented systematic analyses that tie together molecular manipulations with structural and functional analyses.

Here, we combine the strengths of zebrafish for genetic screens, high-resolution live imaging, and electrophysiology to study the growth of axons in diameter and how this growth influences conduction. We identify a mutation in the gene encoding the nuclear transport receptor importin 13b (an ortholog of the gene encoding human Importin 13, a protein highly enriched in the nervous system[42]) that results in a striking reduction in axon diameters. By generating neuron-specific ipo13b mutants and labelling specific neurons to follow their axons over time in vivo, we show that neuronal importin 13b is required for the growth of axons in diameter, but not growth in length, cell body size or thickness of myelin along the axon. Pairing our structural live imaging with electrophysiological recordings from the exact same neuron revealed that conduction velocity along individual myelinated axons increased in relation to increases in diameter, not with developmental age or changes to myelin thickness. However, reducing axon diameter did not affect the precision of action potential propagation or the ability to fire at high frequencies. This implicates diameter growth as a major mechanism driving increases in conduction speed along individual myelinated axons during development.

## Results

### Zebrafish as a model to study axon diameter

To investigate how CNS axons grow in diameter, we used zebrafish as a model system. Within the first five days of development in the zebrafish CNS, there is rapid and differential growth of axons in diameter, such that by 5 days post-fertilisation (dpf), there is already over ~40-fold difference in axon diameter across different neurons (Fig. 1A). In particular, we focussed on the axon diameter growth of the Mauthner neuron, because of its ease of identification, its early and extensive axon diameter growth, and its tractability to perform electrophysiological recordings to assess the association between diameter growth and action potential conduction properties.

The Mauthner neurons are a bilateral pair of reticulospinal neurons within rhombomere 4 of the hindbrain that project their axons down the entire length of the contralateral spinal cord (Fig. 1B, D)[43]. From 2 dpf through to 5 dpf, the Mauthner axon grows rapidly from ~1.0 µm to over 3.5 µm, which can readily be measured using super-resolution confocal live-imaging (Fig. 1C, E and Methods). Myelin forms along the Mauthner axon beginning around 2.5 dpf[44], and by 3.5 dpf it is ensheathed along almost its entire length (Fig. 1F, G), after which myelin continues to grow in thickness as the axon grows in diameter (Fig. 1C). The rapid and extensive diameter growth of the Mauthner axon and its early myelination make it a powerful model to study mechanisms of axon diameter growth, and to determine how manipulating the diameter of an axon affects its myelin and conduction properties.

### Identification of importin 13b mutants with altered axon diameter

Given how little is known about the regulation of axon diameter, unbiased, discovery-based screens are a powerful approach to uncover novel molecular mechanisms. As part of a zebrafish ENU (N-ethyl-N-nitrosourea) mutagenesis-based forward genetic screen aimed at identifying novel regulators of different aspects of myelinated axon development[45–47], we identified a mutant allele, ue57, which resulted in a striking reduction in the diameter of the Mauthner axon in the zebrafish spinal cord (Fig. 2A–D). The ue57 mutant animals exhibited normal overall growth and gross morphology throughout embryonic development and early larval stages (Fig. 2A); however, they displayed altered swim behaviour (decreased frequency and increased duration of swim bouts, Fig. 2E, F) and died between 9 and 11 dpf.

Mapping-by-sequencing of phenotypically mutant larvae localised the causative mutation to a 45–50 MB region on chromosome 20, and revealed a C > T base pair change that introduced a premature stop codon (R924X) within the last exon of the gene ipo13b (ZDB-GENE-070706-1/GRCz11:ENSDARG00000060618), predicted to eliminate the last 30 amino acids of the protein importin 13b (Fig. 2G, H). Importin 13b is an ortholog of the human protein Importin 13 and is well-conserved, with zebrafish importin 13b and human Importin 13 protein sharing 83% amino acid identity. It is a member of the importinβ superfamily, which is a family of receptors responsible for shuttling proteins between the nucleus and the cytoplasm[48,49], including long-range transport roles moving cargoes between the nucleus and the dendrites/axons/synapses of neurons[50–53]. Importin 13 itself is highly expressed within the CNS[42]; however, its specific roles there have yet to be fully explored[54–56].

We confirmed ipo13b was the gene responsible for the ue57 phenotype by ipo13b mRNA rescue (Fig. 2K, L), as well as the generation of two additional mutant alleles (ipo13b^ue76 and ipo13b^ue77). Both mutant alleles introduce frame-shift mutations in a ran-GTP binding site within the N-terminal of the protein that is critical for importin 13 function (Fig. 2I, J)[57]. The ipo13b^ue76 and ipo13b^ue77 mutants, as well as ipo13b^ue57/ue76 and ipo13b^ue57/ue77 trans-heterozygotes, phenocopied the Mauthner axon diameter defect observed in the ue57 mutant (Fig. 2M–P). Thus, we concluded that ipo13b is the gene responsible for the axon diameter phenotypes in the ue57 mutants, and further refer to this mutant line as ipo13b^ue57.

### Importin 13b mutants exhibit reduced CNS axon diameter growth

To determine whether importin 13b is required for axon diameter growth and/or maintenance, we performed super-resolution live-imaging of a fluorescent reporter expressed in the Mauthner neuron to measure its axon diameter over time from 2 to 7 dpf. The time-course analysis revealed that while Mauthner axon diameter is initially normal in homozygous mutants at 2 dpf, growth in diameter significantly slows after this time point compared to both wild-type and heterozygous Mauthner axons (Fig. 3A, B). Despite their smaller diameter, Mauthner axons in mutants still grow to a diameter large enough that they become myelinated (Figs. 2B, C and 4B). Furthermore, we observed no abnormalities in Mauthner axon outgrowth, with axons reaching the same length in both control and ipo13b^ue57 mutant animals (Fig. 3C–E).

To assess whether deficits in axon diameter growth occurred in other types of neurons, we first transgenically labelled and live-imaged individual axons from two additional and easily identifiable pairs of reticulospinal neurons that project axons along the ventral spinal cord, the MiM1 and Mid3i neurons. Both sets of neurons also exhibited significantly reduced axon diameter in ipo13b^ue57 mutants at 5 dpf (Fig. 3F–I). The severity of the phenotype correlated with axon diameter, such that growth of the larger diameter axons was more affected than the smaller diameter axons (growth in diameter reduced

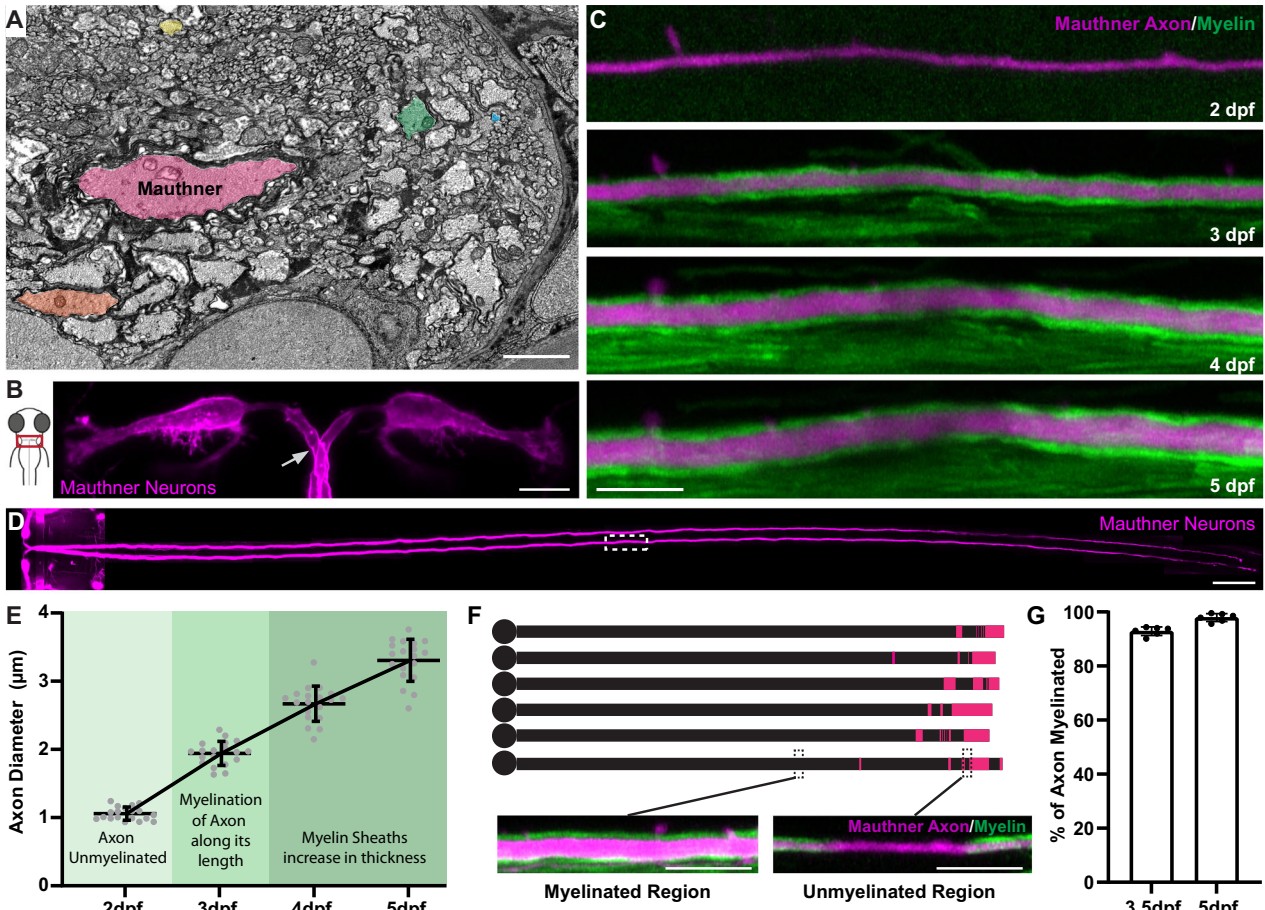

**Fig. 1 | Zebrafish as a model to study axon diameter. A** Electron micrograph of a cross-section of the zebrafish ventral spinal cord at 5 dpf showing the diverse range of axon diameters. Five axons spanning the range of diameters are highlighted: the largest myelinated Mauthner axon (pink), two other myelinated axons (orange and green) and two unmyelinated axons (yellow and blue). **B** Schematic dorsal view of the larval zebrafish head with inset indicating the position of Mauthner neurons shown in right, labelled using the transgenic line Tg(hspGFF62A:Gal4); Tg(UAS:mem-Scarlet). The arrow points to one of the two axons, about to cross the midline. The image was obtained at 5 dpf. **C** Super-resolution confocal live-imaging time-course depicting the growth in diameter of a Mauthner axon (magenta−Tg(hspGFF62A:Gal4); Tg(UAS:mRFP)) with myelination (green−Tg(mbp:eGFP-CAAX)) at somite 15 from 2 to 5 dpf. **D** Tiled dorsal view of the Mauthner neurons (5 dpf), labelled using the transgenic line Tg(hspGFF62A:Gal4); Tg(UAS:mRFP). White boxed region depicts where the time course analysis in **C** was performed (somite

15). **E** Quantification of Mauthner axon diameter growth with relation to its myelination followed for the same axons over time at somite 15 from 2 to 5 dpf ($n = 19$ axons from individual animals). This dataset is the same as the wild-type data shown in Fig. 3B. **F** Schematic of the myelination of six different wild-type Mauthner axons at 3.5 dpf, with myelinated regions represented by black and unmyelinated regions represented by magenta, which is quantified in (**G**). Insets show a myelinated and unmyelinated region of the axon with the axon labelled using the transgenic lines Tg(hspGFF62A:Gal4); Tg(UAS:mRFP) and myelin labelled using the transgenic line Tg(mbp:eGFP-CAAX). **G** Quantification of the percentage of the Mauthner axon myelinated at 3.5 dpf and 5 dpf ($n = 6$ axons from individual animals). All graphs are presented as mean values ± SD, with repeated measures for the same axon at each time point. Scale bars: 1 μm (**A**), 10 μm (**B**, **C**, **F**), 100 μm (**D**). Source data are provided as a Source Data file.

by 48% for Mauthner axons, 39% for MiM1 axons, and 13% for Mid3i axons at 5 dpf).

Next, we assessed axon diameter in cross sections of the 7 dpf zebrafish spinal cord using electron microscopy (Fig. 4A, B). We measured a 51% reduction in Mauthner axon diameter (80% reduction by area), which was comparable to the 55% reduction observed at 7 dpf using live-imaging (Fig. 4C vs. 3B). The average axon diameter of the next 30 largest axons per ventral spinal cord hemisegment was also significantly reduced, with binning of these axons based on their axon diameter showing a global shift to smaller diameters in *ipo13b^ue57* mutants (Fig. 4D, H). There was also a small shift towards smaller axon diameters amongst the 30 largest axons in the dorsal spinal cord (Fig. 4E, F, I). As dorsal axons are on average smaller in diameter than ventral axons, the smaller effect on dorsal axons is in agreement with our live-imaging results indicating that larger diameter axons are more significantly affected by loss of importin 13b function. Our electron microscopy analyses also

revealed that fewer axons were ensheathed with myelin in the spinal cord of *ipo13b^ue57* mutants compared to their sibling controls (Fig. 4G). Given the role of axon diameter in the selection of axons that become myelinated[4–8], the reduction in myelin is likely at least in part a consequence of the reduction in axon diameter. Together, our live-imaging and electron microscopy studies support the conclusion that importin 13b is required for the proper growth of axon diameter for a wide range of neurons.

### Disruption of importin 13b function affects both neurofilament number and spacing

To better understand how the loss of importin 13b function affects axon diameter, we looked at the axonal cytoskeleton within *ipo13b^ue57* mutant Mauthner axons. In general, the growth of myelinated axons in diameter is associated with an increased number of neurofilaments and/or reduced neurofilament density (increased spacing between neurofilaments) (reviewed in[58,59]). We

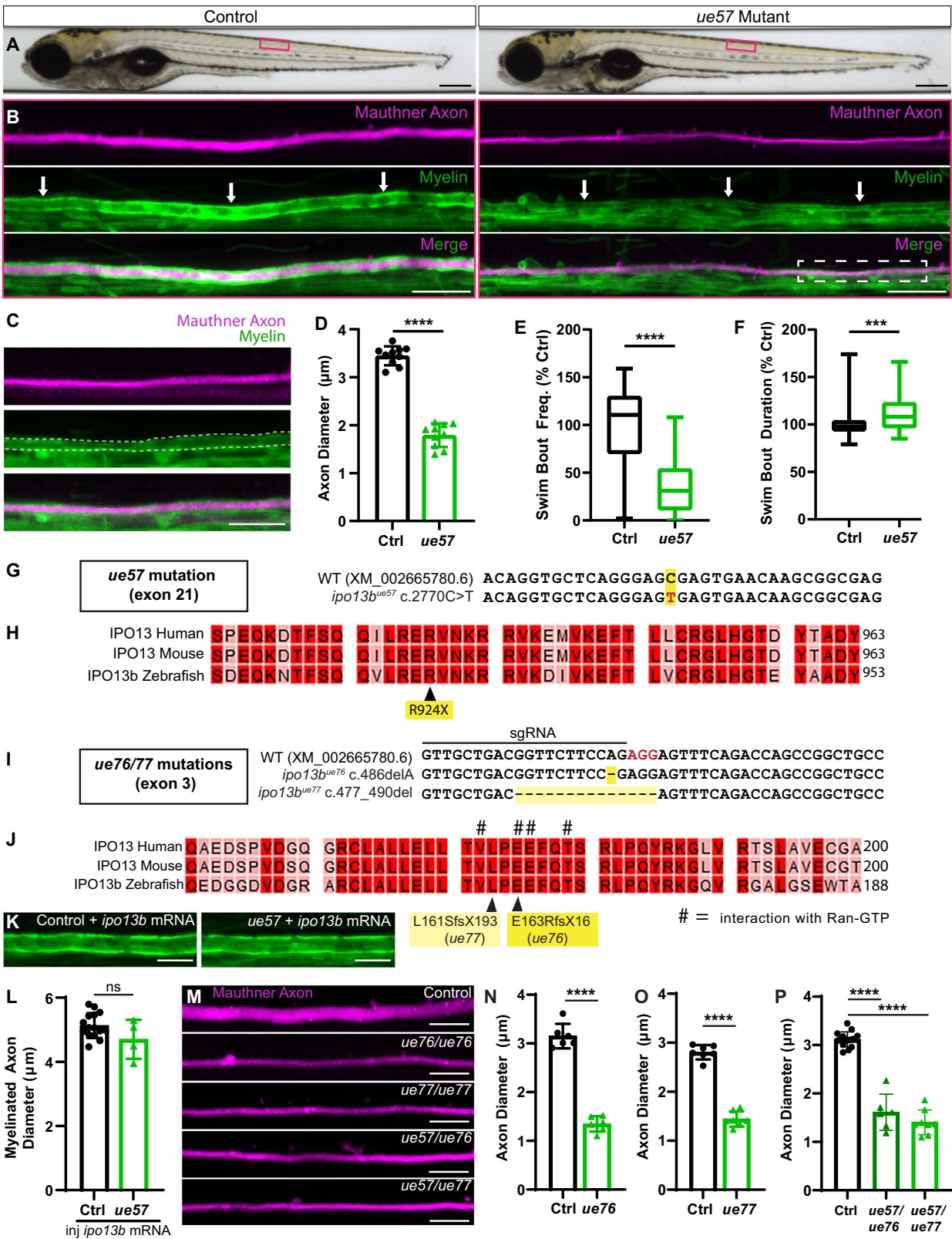

analysed the density of neurofilaments in cross sections of 7 dpf control and *ipo13b*[ue57] mutant Mauthner axons in electron microscopy images (Fig. 5A–D) and found that there was a 29% increase in neurofilament density in *ipo13b*[ue57] mutant Mauthner axons (Fig. 5E). Likewise, nearest neighbour analysis revealed a shift towards shorter distances between individual neurofilaments (Fig. 5F), indicating they are spaced more closely together.

However, given the roughly 80% reduction in the cross-sectional area of *ipo13b*[ue57] mutant Mauthner axons, one would have expected a 4–5× increase in neurofilament density if the number of neurofilaments was unchanged, indicating that the total number of neurofilaments in the axon is also reduced. Thus, both the number and density of neurofilaments are affected by disruption to importin 13b function.

**Fig. 2 | Identification of importin 13b mutants with reduced axon diameter growth. A** Brightfield images of control and *ue57* mutant zebrafish at 5 dpf, depicting normal growth and gross morphology. **B** The Mauthner axon (somite 15) labelled using Tg(hspGFF62A:Gal4); Tg(UAS:mRFP) in control (left panels) and *ue57* mutant (right panels) is of smaller diameter in mutants at 5 dpf. Labelling of myelin with Tg(mbp:eGFP-CAAX) shows that the Mauthner axon is myelinated (white arrows). Magnification of the mutant Mauthner axon (region within the dashed rectangle) is shown in (**C**) with the myelin along the Mauthner axon outlined with dashed lines. **D** Mauthner axon diameter measured at somite 15 using the Tg(hspGFF62A:Gal4); Tg(UAS:mRFP) reporter at 5 dpf (two-tailed unpaired *t*-test, *n* = 10 axons from individual animals per genotype, *p* < 0.0001). **E** In a 20 min open field test at 5 dpf, *ue57* mutant zebrafish initiate fewer swim bouts than controls; however, swim bouts are slightly longer in length (**F**) (two-tailed Mann–Whitney test, *p* < 0.0001 for (**E**) and *p* = 0.001 for (**F**), *n* = 144 control and 31 *ue57* animals). **G** A region of exon 21 (last exon) of the *ipo13b* gene where a C > T base pair change (highlighted in red) was identified in *ue57* mutants. **H** This base pair change results in the introduction of a premature stop codon in the highly conserved C-terminal region of importin 13b, predicted to result in a truncated protein missing the last 30 amino acids. **I** Overview of a region in exon 3 of the *ipo13b* gene indicating the site targeted with a sgRNA for Cas9-mediated DNA cleavage (PAM sequence highlighted in red), and the resulting mutations in the *ue76* and *ue77* mutant lines. **J** The mutations disrupt key residues previously shown to bind Ran-GTP (marked with #)[57], and result in frame shifts followed by premature stop codons. **K** Representative images of the myelinated Mauthner axon (somite 15, 5 dpf) labelled using Tg(mbp:eGFP-CAAX) in control and *ue57* zebrafish that were injected with 125 pg of *ipo13b* mRNA at the single cell stage, with quantification in (**L**) showing rescue of the axon diameter phenotype (two-tailed unpaired *t*-test, *n* = 15 control and 4 *ue57* axons from individual animals, *p* = 0.096). **M** Representative images of the Mauthner axon (somite 15) labelled using the Tg(hspGFF62A:Gal4); Tg(UAS:GFP) reporter in 4–5 dpf control, *ipo13b*<sup>*ue76*</sup>, *ipo13b*<sup>*ue77*</sup>, *ipo13b*<sup>*ue57/ue77*</sup>, *ipo13b*<sup>*ue57/ue77*</sup> zebrafish, with quantification of axon diameter compared to control siblings shown in (**N**–**P**) (**N**—two-tailed unpaired *t*-test, *n* = 6 axons from individual animals per genotype, *p* < 0.0001; **O**—two-tailed unpaired *t*-test, *n* = 7 axons from individual animals per genotype, *p* < 0.0001, **P**—One-way ANOVA with Tukey's multiple comparisons test, *n* = 15 control, 6 *ue57/ue76*, and 8 *ue57/ue77* axons from individual animals, *p* < 0.0001)). Data are presented as mean values ± SD, except **E** and **F** where data is presented as box and whisker plots with boxes indicating the median, 25th and 75th percentiles, and whiskers indicating the max and min. ***p* < 0.001, ****p* < 0.0001, ns = not significant. Scale bars: 300 μm (**A**), 20 μm (**B**), 10 μm (**C**, **K**, **M**). Source data are provided as a Source Data file.

## Axon diameter growth requires importin 13b function in neurons

Importin 13 is expressed widely throughout different tissues, including in all major CNS cell types[60]. Thus, importin 13b may be required within neurons for their axons to grow in diameter, and/or it may play a role in non-neuronal cells and exert its effect on axon diameter in a non-cell autonomous manner[9,61–65]. For example, in the peripheral nervous system, signals from myelinating Schwann cells can both promote and restrict axon diameter growth[62,63,65]. To investigate the role of importin 13b specifically in neurons, we generated neuron-specific *ipo13b* mutants using a cell-type specific CRISPR/Cas9 strategy (Fig. 6A, Methods). Neuron-specific *ipo13b* mutant fish live a normal life span; however, similar to the *ipo13b*<sup>*ue57*</sup> mutants, they have Mauthner axons with reduced axon diameter without affecting axon length (Fig. 6B–D). We also assessed the growth of the cell body and lateral dendrite in relation to axon diameter growth (Fig. 6E). Unlike the reduced axon diameter growth observed between 3 and 5 dpf, there was no significant difference in the size of the cell body and lateral dendrite for control and neuron-specific *ipo13b* Mauthner neurons, which grew significantly between these timepoints (Fig. 6F–H). Together, these data indicate that neuronal *ipo13b* influences axon diameter growth independently of axon outgrowth or overall growth of the neuron.

Notably, approximately 20% of fish expressing both neuronal Cas9 and sgRNAs targeting *ipo13b* showed no observable defects in the diameter of one or both of their Mauthner axons (Fig. 6B). Given the mosaic nature of the mutations introduced using the CRISPR/Cas9 strategy, we predict that this reflects the introduction of in-frame or silent *ipo13b* mutations within these neurons that do not lead to bi-allelic loss-of-function. For this reason, fish positive for both the Cas9 and sgRNA transgenes, but showing no reduction in axon diameter when compared to control siblings, were excluded from further analysis. It was also notable that for affected Mauthner axons, the reduction in axon diameter was not as severe as in full *ipo13b* mutants (48% reduction at 5 dpf for *ipo13b*<sup>*ue57*</sup> mutants vs. 29% reduction at 5 dpf for neuron-specific *ipo13b* mutants). This may be due to the timing of *ipo13b* disruption (e.g. some functional importin 13b is expressed in the cells before the gene is disrupted), or additional non-cell-autonomous roles for importin 13b in regulating axon diameter. Nonetheless, the reduction in axon diameter in the neuron-specific *ipo13b* mutants provides a model with which to test how axon diameter growth affects conduction along individual axons without the confounding effects from importin 13b function in other cells, such as myelinating oligodendrocytes.

## Myelin thickness is unchanged along smaller diameter *ipo13b* mutant Mauthner axons

Axon diameter influences conduction along the axon both directly, by reducing the axial resistance to ion flow, and indirectly, by affecting myelin. Previous studies comparing axons of different diameters have shown that there is a positive correlation between axon diameter and myelin thickness[12,13,66] and that growth of axons in diameter is generally associated with increased thickness of myelin sheaths[19,63]. However, other studies have shown that alterations in axon diameter do not necessarily lead to proportional changes to myelin thickness[36,67]. Therefore, it remains unclear the extent to which axon diameter directly regulates the growth of myelin in thickness. Indeed, particularly little is known about how myelin actually changes over time along individual axons as they grow in diameter during development or how reducing axon diameter growth affects the amount of myelin made around the axon.

Using time course live-imaging to follow the same axon over time, we assessed how myelin thickness changed along control Mauthner axons as they grew in diameter and found that myelin thickness increased significantly between 3 dpf and 5 dpf (Fig. 7A, C, D). To further understand the relationship between axon diameter and myelin thickness over time, we calculated the g-ratio, which is the ratio of the axon diameter relative to the diameter of the axon plus the myelin sheath (Fig. 7B, Supplemental Fig. 1). The g-ratio is a parameter predicted to impact conduction speeds along myelinated axons and in general is found to be fairly constant for axons of the same diameter[13,68,69]. We found as the Mauthner axon grew in diameter that its g-ratio changed only slightly, with a small but significant decrease between 3 and 5 dpf (Fig. 7E).

Next, we asked how the thickness of myelin on the Mauthner axon changed when its diameter growth was reduced. The Mauthner axons in neuron-specific *ipo13b* mutant animals did grow to diameters that exceeded the threshold for myelination and were fully myelinated along their length (Fig. 7F). In addition, we found that myelin thickness was comparable between control and neuron-specific *ipo13b* mutant Mauthner axons at 5 dpf, despite the mutant axons being smaller in diameter (Fig. 7G–I, K–M, Supplemental Fig. 2). This would suggest that the growth of myelin in thickness along the Mauthner axon over this time period was not in direct response to the change of the axon in diameter (see discussion). The fact that myelin thickness was unchanged, but axon diameter was decreased meant that the g-ratio was significantly smaller for neuron-specific *ipo13b* mutant Mauthner axons compared to controls (Fig. 7J, N). This provided us with the opportunity to assess how reducing axon diameter, without changing myelin thickness, affects the maturation of conduction properties of single myelinated axons.

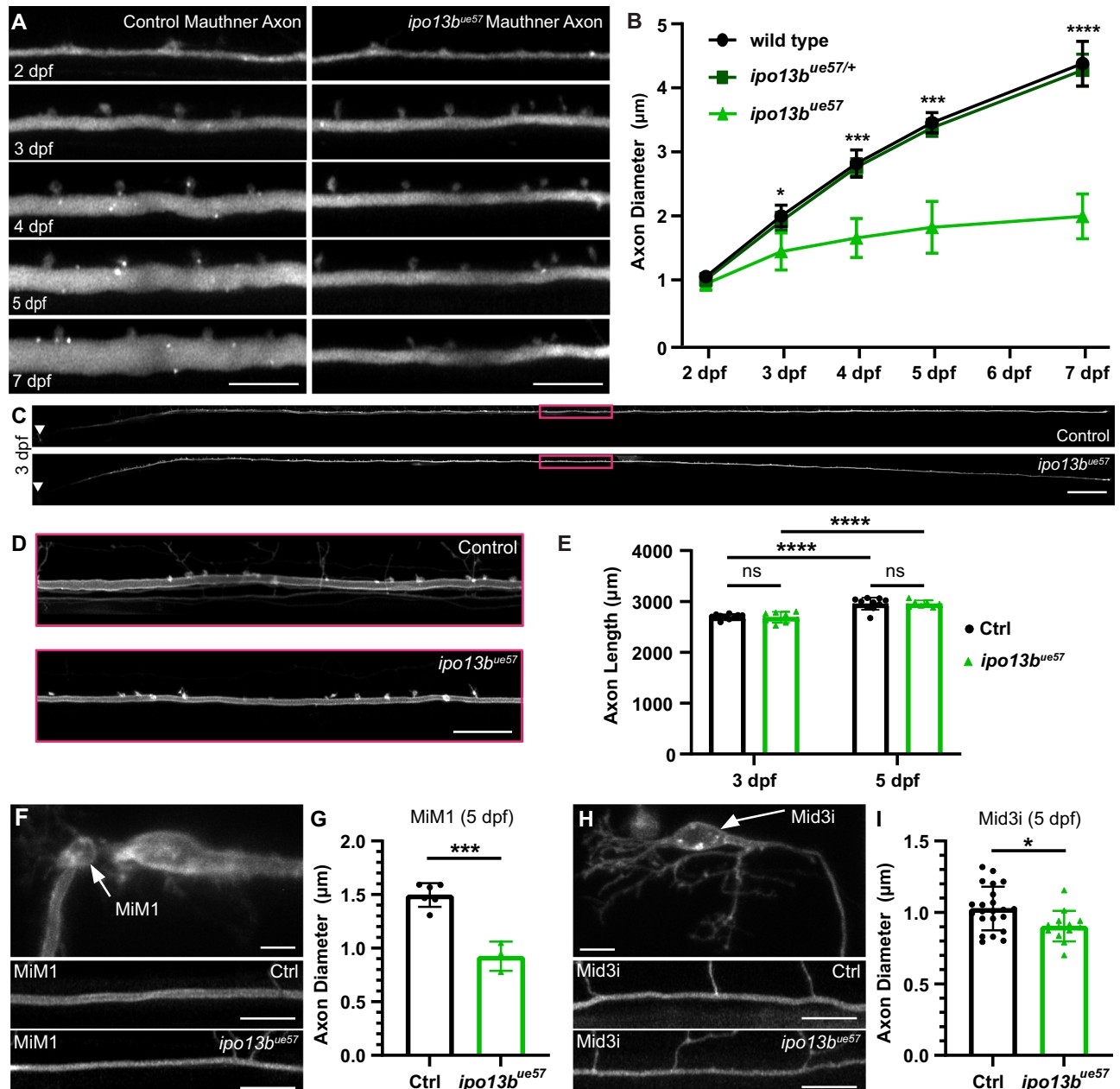

**Fig. 3 | Live-imaging of axon diameter growth defects in importin 13b mutants.**
**A** Representative live-imaging time course of the Mauthner axon (somite 15) from 2 to 7 dpf in a control and *ipo13b*[ue57] zebrafish labelled using Tg(hspGFF62A:Gal4); Tg(UAS:GFP). **B** Quantification of Mauthner axon diameter growth followed for the same axons at somite 15 from 2 to 7 dpf (2-way RM ANOVA with Tukey's multiple comparisons test, $n = 19$ wild type, 33 heterozygous, 6 mutant axons from individual animals with repeated measures at each time point, $p = 0.016$ (3dpf), $p = 0.0003$ (4dpf), $p = 0.0002$ (5dpf), $p < 0.0001$ (7dpf) for wild type vs. mutant comparisons, wild type vs. heterozygous are not significantly different from one another). **C** Representative live images of the entire Mauthner neuron in control and *ipo13b*[ue57] zebrafish at 3 dpf labelled using Tg(hspGFF62A:Gal4); Tg(UAS:mem-Scarlet), which were used to measure axon length in (**E**). Area boxed in magenta is enlarged in (**D**). **E** Quantification of the entire length of the Mauthner axon at 3 and 5

dpf (two-way RM ANOVA with Uncorrected Fisher's LSD, $n = 10$ control and 7 *ipo13b*[ue57] axons from individual animals with repeated measures at each time point, $p > 0.0001$ for 3 dpf vs. 5 dpf comparisons, no significant differences between genotypes at 3 dpf ($p = 0.842$) or 5 dpf ($p = 0.996$)). **F** MiM1 and (**H**) Mid3i neurons (arrows in top panels) and their axons (somite 15, 5 dpf) in control (middle panels) and *ipo13b*[ue57] (bottom panels) animals labelled using the transgenic reporter Tg(hspGFF62A:Gal4); Tg(UAS:mem-Scarlet). **G** Quantification of MiM1 axon diameter at 5 dpf (two-tailed unpaired *t*-test, $n = 6$ control and 3 *ipo13b*[ue57] axons from individual animals, $p = 0.0003$). **I** Quantification of Mid3i axon diameter at 5 dpf (two-tailed unpaired *t*-test, $n = 22$ control and 13 *ipo13b*[ue57] axons from individual animals, $p = 0.016$). All data are presented as mean values ± SD. *$p < 0.05$, ***$p < 0.001$, ****$p < 0.0001$, ns = not significant. Scale bars: 10 μm (**A**, **F**, **H**), 100 μm (**C**), 20 μm (**D**). Source data are provided as a Source Data file.

## The growth of axons in diameter drives dynamic changes to conduction speed along myelinated axons

To study how axon diameter growth relates to conduction property maturation, we took advantage of the capacity to both live-image individual Mauthner axons and record the electrophysiological properties of the exact same neuron in intact zebrafish larvae (Fig. 8A–C).

Given the well-established association between axon diameter and conduction speed, we first examined the relationship between axon diameter and the velocity of action potential propagation along individual Mauthner axons as they grew in diameter over time. To begin, we measured both axon diameter and conduction velocity along control Mauthner axons between 3 and 5 dpf, a period during which

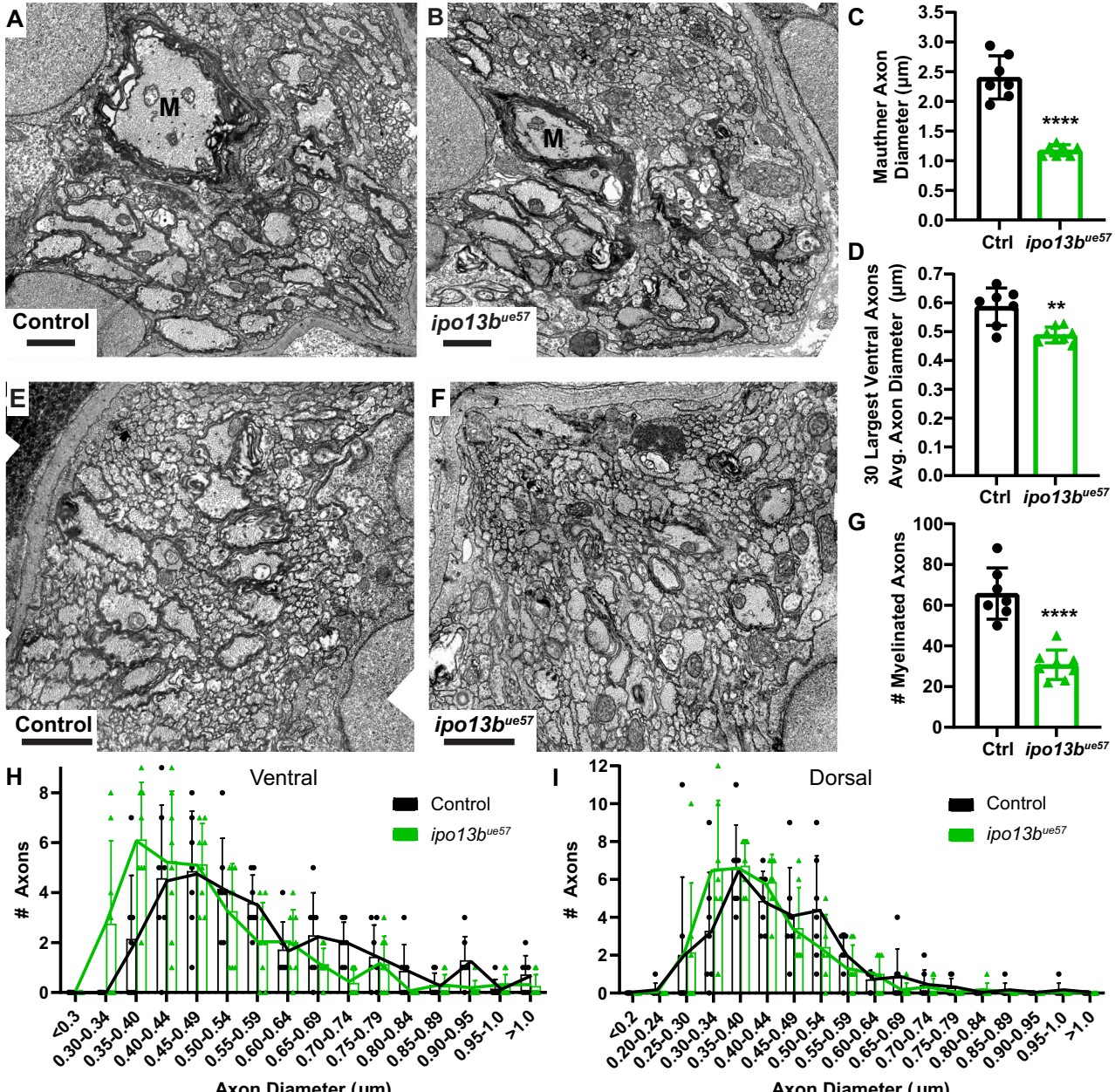

**Fig. 4 | Electron microscopy of axon diameter growth defects in importin 13b mutants. A** Representative electron micrographs of cross sections of the ventral spinal cord at 7 dpf in control and **B** *ipo13b*<sup>ue57</sup> animals. The Mauthner axon is labelled 'M'. **C** Quantification of Mauthner axon diameter from 7 dpf electron micrographs (two-tailed unpaired *t*-test with Welch's correction, *n* = 7 axons from individual animals, *p* < 0.0001). **D** Mean diameter for the 30 largest axons in each hemi ventral spinal cord at 7 dpf, excluding Mauthner (two-tailed unpaired *t*-test with Welch's correction, *n* = 7 control and 8 *ipo13b*<sup>ue57</sup> animals, *p* = 0.006). **E** Representative electron micrographs of cross sections of the dorsal spinal cord at

7 dpf in control and **F** *ipo13b*<sup>ue57</sup> zebrafish. **G** Number of myelinated axons in the dorsal and ventral tracts per hemi spinal cord at 7 dpf (two-tailed unpaired *t*-test, *n* = 7 control and 8 *ipo13b*<sup>ue57</sup> animals, *p* < 0.0001). **H** Distribution of axon diameters for the 30 largest axons in each hemi ventral spinal cord at 7 dpf, excluding Mauthner (*n* = 7 control and 8 *ipo13b*<sup>ue57</sup> animals). **I** Distribution of axon diameters for the 30 largest axons in each hemi dorsal spinal cord at 7 dpf (*n* = 7 control and 8 *ipo13b*<sup>ue57</sup> animals). All data are presented as mean values ± SD. **\*\****p* < 0.01, **\*\*\*\****p* < 0.0001. Scale bars = 1 μm. Source data are provided as a Source Data file.

the axon is already myelinated and continues to grow in diameter (Fig. 1C, E, G). As would be predicted from previous studies comparing myelinated axons of different diameters[10,11], we found that conduction velocity increased in a linear relationship with the growth of the Mauthner axon in diameter over time (Fig. 8F).

To assess the contribution of axon diameter growth to these changes in conduction speed, we next assessed conduction velocity along Mauthner axons in 5 dpf neuron-specific *ipo13b* mutants, which have axons with similar diameters to control axons at 3 dpf. If axon diameter was the main driver of the observed increases in

conduction speed along the myelinated Mauthner axon over time, one would expect the relationship between axon diameter and conduction speed to be maintained, such that axons with the same diameter would have the same conduction speeds regardless of developmental age. In contrast, if the ongoing growth of myelin in thickness or changes to other parameters along the axon contributed significantly to changes in conduction speed over time, one would expect that neuron-specific *ipo13b* mutant Mauthner axons would conduct action potentials faster than the younger control axons of equivalent axon diameter.

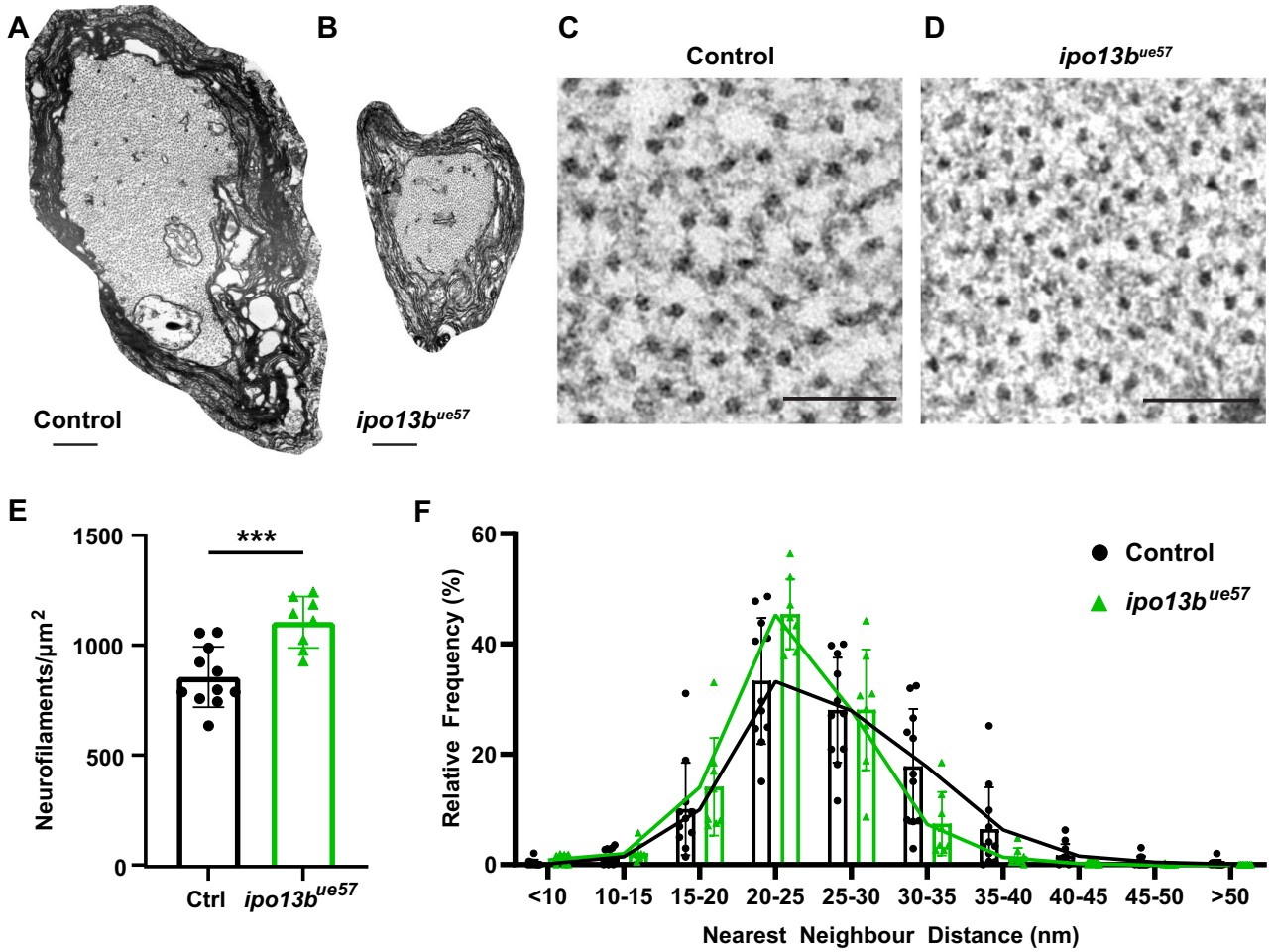

**Fig. 5 | Disruption of importin 13b function increases neurofilament density in the Mauthner axon. A, B** Representative electron micrographs of a cross-section of the Mauthner axon in the ventral spinal cord at 7 dpf in control and *ipo13b*[ue57] zebrafish. A region of each axon is magnified in (**C**) (control) and (**D**) (*ipo13b*[ue57]) to allow visualisation of the distribution of neurofilaments. **E** Density of neurofilaments in control and *ipo13b*[ue57] mutant Mauthner axons at 7 dpf (two-tailed unpaired *t*-test, *n* = 11 control and 8 *ipo13b*[ue57] animals, *p* = 0.0007). **F** Nearest neighbour distribution of neurofilaments in the Mauthner axon at 7 dpf, showing a shift to closer nearest neighbours in the *ipo13b*[ue57] mutants (*n* = 11 control and 8 *ipo13b*[ue57] animals). Scale bars: 500 nm (**A, B**), 100 nm (**C, D**). All data are presented as mean values ± SD. \*\*\**p* < 0.001. Source data are provided as a Source Data file.

As would be predicted for their reduced axon diameter, we found that neuron-specific *ipo13b* mutant Mauthner axons did conduct action potentials slower than control axons of the same development stage (Fig. 8D, E). Strikingly however, the relationship between axon diameter and conduction speed was maintained, with conduction speeds along mutant axons comparable to control axons of the same axon diameter at earlier developmental stages (Fig. 8F). Together, these data indicate that the developmental changes to conduction velocity observed along the myelinated Mauthner axon over time are tightly linked to its growth in diameter.

**Reduced axon diameter does not affect the ability to fire at high frequencies or the precision of action potential firing**
In addition to its influence on conduction velocity, it has also been speculated that larger diameter axons better support higher frequency firing of action potentials[3], but it has not been tested how changes to the diameter of an individual axon might affect its ability to support the firing of action potentials at different rates. Therefore, we asked whether the ability of the Mauthner axon to sustain high-frequency firing was influenced by its diameter. To test this, we examined the action potential activity evoked by a range of depolarising stimuli delivered at 300 Hz, 500 Hz and 1000 Hz to Mauthner axons in control and neuron-specific *ipo13b* mutant animals. Despite being smaller in

diameter than controls, neuron-specific *ipo13b* mutant axons maintained even the highest frequency trains of stimuli (Fig. 8H). This counters the view that axon diameter is a parameter that is specifically regulated to sustain high-frequency firing patterns, at least along the myelinated axons examined here. Interestingly, we recently reported that hypomyelination of the Mauthner axon does affect its ability to fire at high frequencies[70]. Thus, together these results would suggest that it is the myelin on large-diameter axons that is important to support high-frequency firing rather than their diameter itself.

We also assessed the impact of axon diameter on the precision of action potential propagation, which is important for the coordinated and synchronised release of presynaptic vesicles and subsequent synaptic transmission. This was measured as "jitter," which was calculated as the standard deviation in the timing of action potential arrival for 30 stimulated action potentials (Fig. 8I). No difference in the precision of action potential arrival was observed along Mauthner as it grew in diameter over time in controls, or in neuron-specific *ipo13b* mutants with reduced axon diameter (Fig. 8J). Furthermore, we also did not detect any differences in the resting membrane potential of the neuron-specific *ipo13b* mutant Mauthner neurons indicating that they have normal intrinsic excitability (Fig. 8G). Together, our data indicates disrupting axon diameter growth specifically affects the speed of action potential propagation, without affecting intrinsic excitability,

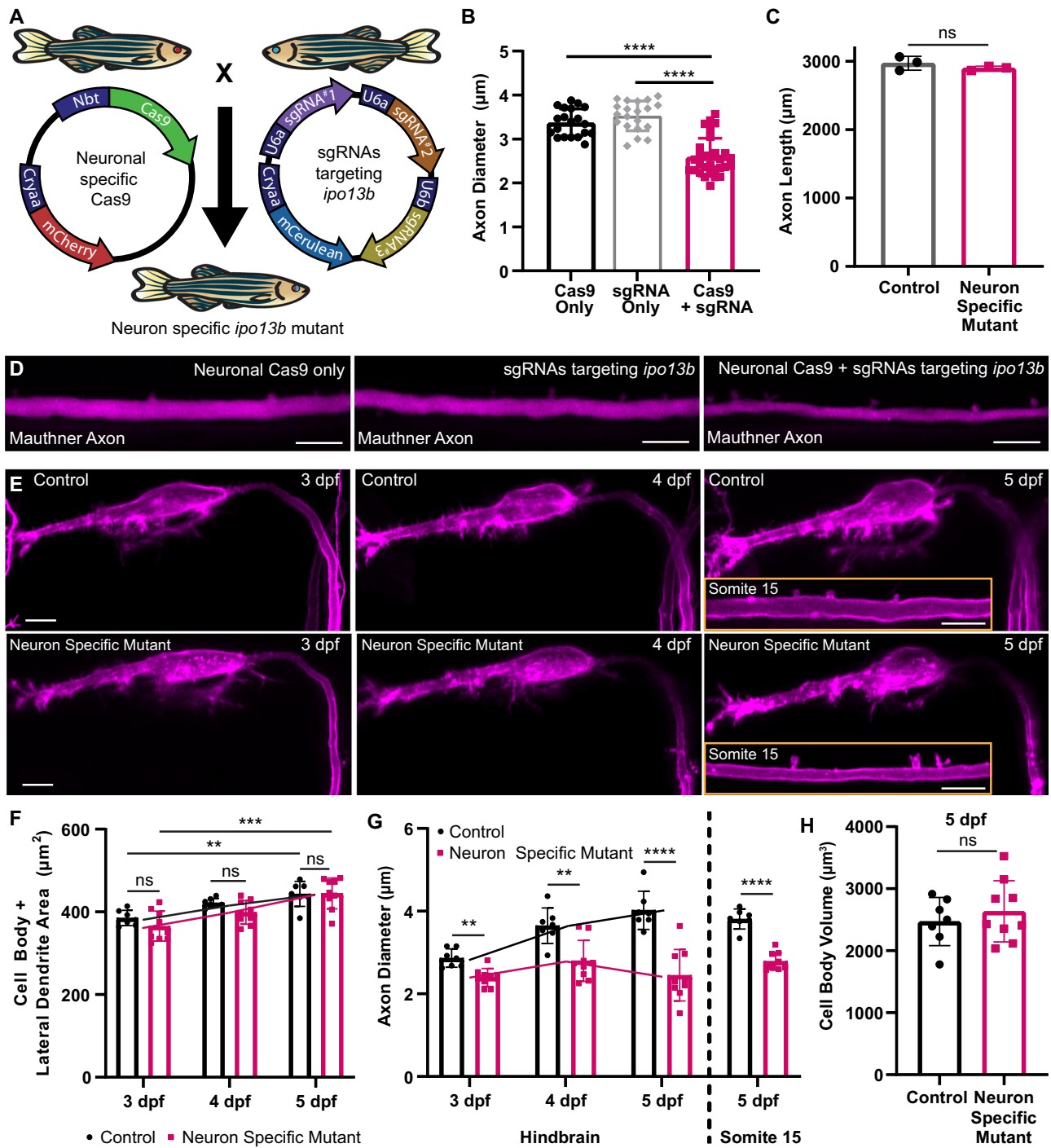

the ability to conduct action potential at high frequencies, or the precision of action potential conduction.

## Discussion

Here we exploited the tractability of zebrafish by combining genetics, in vivo imaging, and electrophysiology to study how axons grow in diameter and how axon diameter growth affects the conduction properties of individual myelinated axons over time in vivo. We demonstrate that the nuclear transport receptor importin 13b is essential for axon diameter growth via a mechanism that affects both neurofilament number and density but does not influence axon outgrowth or growth of the cell body. In neuron-specific *ipo13b* mutants, the reduction in axon diameter growth did not change the thickness of myelin around the axon. By combining super-resolution live imaging

with electrophysiology, we show that axon diameter growth drives increases in the conduction speed of myelinated axons, but is not required to sustain high-frequency action potential propagation or for the precision of action potential conduction. Together, this highlights the involvement of importin 13b in a neuronal mechanism of growth that is specific to axon diameter, which regulates increases in conduction speeds along myelinated axons over time.

An advantage of our live-imaging paradigm is the ability to image single neurons in their entirety, allowing us to assess not only changes to axon diameter over time, but also how this relates to other aspects of neuronal growth. Many previous manipulations of axon diameter in other systems have relied on targeting global growth pathways within the neuron (e.g. AKT pathway) that modulate multiple aspects of neuronal growth, not just growth in axon diameter[5,36–38]. However, here

**Fig. 6 | Disruption of axon diameter in neuron-specific *ipo13b* mutants.**
**A** Schematic overview of the transgenic CRISPR/Cas9 strategy used to generate neuron-specific *ipo13b* mutants. **B** Quantification of Mauthner axon diameter in control (Cas9 or sgRNA only) and neuron-specific *ipo13b* mutants (Cas9+sgRNA) at 5 dpf (Kruskal–Wallis test with Dunn's multiple comparisons test, *n* = 20 Cas9 only, 20 sgRNA only, 30 Cas9 + sgRNA axons (1–2 axons per animal), *p* < 0.0001). **C** Quantification of Mauthner axon length in control and neuron-specific *ipo13b* mutants at 5 dpf (two-tailed unpaired *t*-test, *n* = 3 axons from individual animals for each genotype, *p* = 0.2943). **D** Representative images of the Mauthner axon (somite 15, 5 dpf) in control and neuron-specific *ipo13b* mutant animals labelled using Tg(hspGFF62A:Gal4); Tg(UAS:mRFP). **E** Representative time course images of the Mauthner cell body, dendrites, and proximal axon in control and neuron-specific *ipo13b* mutants from 3 to 5 dpf labelled using Tg(hspGFF62A:Gal4); Tg(UAS:mem-Scarlet). The inset at 5 dpf shows the axon at somite 15. **F** Quantification of the area of the Mauthner cell body and lateral dendrite measured for the same neurons from 3 to 5 dpf (two way RM ANOVA with Tukey's multiple comparisons tests, *n* = 7 control and 9 neuron-specific mutant Mauthner cells from individual animals with repeated measures at each time point, *p* = 0.003 (control 3 dpf vs. 5 dpf) and *p* = 0.0004 (neuron-specific mutant 3 dpf vs 5 dpf); control vs. neuron-specific mutants are not significant at any time points *p* = 0.1834 (3 dpf), *p* = 0.0504 (4 dpf) and *p* = 0.9443 (5 dpf). **G** Quantification of Mauthner axon diameter (proximal region of the axon located in the hindbrain) measured from 3 to 5 dpf for the same neurons as in (**F**) (two-way RM ANOVA with Tukey's multiple comparisons tests, *n* = 7 control and 9 neuron-specific mutant Mauthner cells from individual animals with repeated measures at each time point, *p* = 0.0010 (3 dpf), *p* = 0.0026 (4 dpf), and *p* < 0.0001 (5 dpf)). Measurements of axon diameter at somite 15 for the same axons are also shown for the 5 dpf time point (two-tailed unpaired *t*-test, *n* = 7 control and 9 neuron-specific mutant Mauthner cells from individual animals, *p* < 0.0001). **H** Quantification of the volume of the Mauthner cell body at 5 dpf (two-tailed unpaired *t*-test, *n* = 7 control and 9 neuron-specific mutant Mauthner cells from individual animals, *p* = 0.482). All data are presented as mean values ± SD. \*\**p* < 0.01, \*\*\**p* < 0.001, \*\*\*\**p* < 0.0001, ns = not significant. Scale bars = 10 μm. Source data are provided as a Source Data file.

we show that importin 13b mutant neurons have axons with smaller diameters, but normal axon length and cell body size. It is worth noting that the specificity of importin 13b function for diameter growth, and not growth of cell body, was assessed in neuron-specific *ipo13b* mutants. In line with the widespread expression of *ipo13* [71] and the lethal nature of the full mutant (lifespan ~9 days), we observed signs of neurodegeneration in a proportion of *ipo13b^ue57* fish, including dying back of axons, retraction of dendrites, and/or rounding of cell bodies, which affected ~25% of Mauthner neurons by 7 dpf. Thus, complete loss of importin 13b likely has additional effects on neurons, which could provide the basis for future studies.

Importin 13b offers a molecular entry point to studying the specific control of axon diameter and how this influences conduction along myelinated axons. We found that axon diameter growth requires importin 13b expressed by neurons and that disruption of importin 13b results in axons with fewer neurofilaments that are also more densely packed within the axon. Neurofilaments are the major cytoskeletal component that supports the growth of large-diameter axons, but the molecular mechanisms controlling their expression and their post-translational modifications that influence diameter remain poorly understood. Therefore, extensive future studies will be required to uncover the molecular pathway(s) by which importin 13b functions to impact neurofilaments and axon diameter. Over 600 candidate cargo for importin 13 have been identified, ranging from transcription factors controlling gene transcription to proteins that directly contribute to or modify the axonal cytoskeleton[72–75]. Whether importin 13b and its cargo function as part of a cell-intrinsic diameter growth pathway and/or in a pathway that responds to cell-extrinsic signals that regulate diameter also remains unclear. For example, importin 13 has been shown to localise at the synapse in Drosophila[54], raising the intriguing possibility that importin 13b could be involved in the transport of activity-related or target-derived signals that might regulate the growth of axons in diameter[9,18–20]. Furthermore, while we show that importin 13b functions within neurons to affect diameter growth, we have not assessed whether this role is mediated entirely by a cell-autonomous requirement for *ipo13b* or might also be influenced by its function in upstream or target neurons. Targeted gene disruption or rescue of *ipo13b* in specific neurons within well-defined circuits would be helpful to address this, which is becoming ever more tractable due to the development of improved techniques in zebrafish that can disrupt gene function in a cell-type and temporal-specific manner without the mosaicism inherent to the CRISPR/Cas9 based cell-type specific strategy used in this study[76,77].

While axon diameter is an important parameter controlling the speed of action potential propagation along myelinated axons, conduction velocity is also influenced by several other factors, including myelin thickness/g-ratio, the distance between and dimensions of nodes of Ranvier, and the types and density of ion channels along the axon[10,14]. Many of these parameters change in parallel as myelinated axons grow in diameter over time in vivo. For example, as myelinated axons grow in diameter their myelin sheaths will often increase in thickness, and it is thought that this is at least in part regulated by axon diameter itself[19,63,78]. However, other studies have reported that changes to myelin thickness do not scale with changes to axon diameter, which in some cases may reflect homoeostatic compensation to maintain the appropriate conduction speed[36,67]. We found for the Mauthner axon that there was a disconnect between its diameter growth and increases in myelin thickness along the axon, such that myelin was the same thickness along control and neuron-specific *ipo13b* mutant axons, despite differences in their axon diameter. Instead, myelin thickness appeared tightly linked to the location along the length of the axon (Supplementary Fig. 2), reflecting the temporal pattern of myelination, which begins along the proximal regions of the axon and then proceeds distally. These data indicate that myelin thickness may primarily reflect the developmental age of the myelin sheath and the default propensity of oligodendrocytes to differentiate and form myelin[4,6,79,80]. Alternatively, the growth of the myelin sheath in thickness may be tuned to axon diameter at the time of initial ensheathment (at which time axon diameter would have been similar for control and *ipo13b* mutant Mauthner axons) rather than responding to ongoing changes in diameter, with differences in myelin thickness along the length of the axon reflecting the tapering of axon diameter along the proximal to distal length. A more detailed understanding of the mechanisms that contribute to the regulation of both axon diameter and myelination is necessary to better understand how typical g-ratios are established for CNS axons.

Across myelinated axons of different diameters, studies have shown that there is a linear relationship between conduction velocity and axon diameter; however, in the context of an individual axon, little is known about the extent to which changes to axon diameter contribute to changes in conduction properties. We took advantage of the fact that we could compare conduction along myelinated axons of the same neuron type, at the same development time point, and with the same myelin thickness, but with significantly different axon diameters. Interestingly, we found that conduction velocity along smaller diameter Mauthner axons in neuron-specific *ipo13b* mutants was precisely scaled to their diameter, such that they had the same conduction velocity observed along younger control axons of the same diameter. This suggests that axon diameter growth is what drives the increases in conduction speed observed over time. The diameter-driven increases in conduction speed may be solely due to diameter itself or also due to changes in other parameters that are driven by, or co-regulated with, diameter growth. Therefore, it will be important to explore whether other features of neuronal development (for example, the transport or

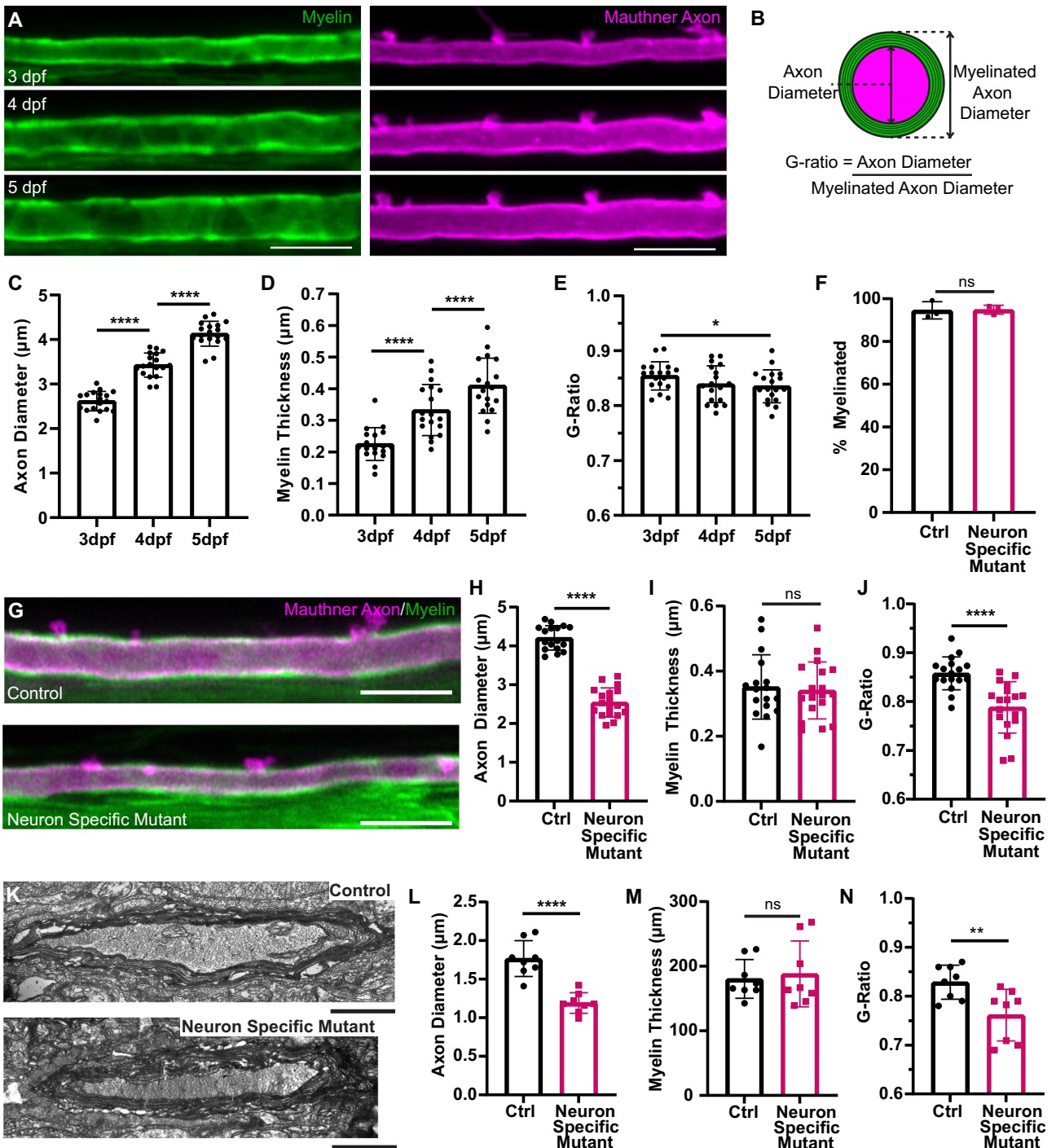

**Fig. 7 | Reduced axon diameter growth does not change myelin thickness along the Mauthner axon. A** Representative time course from 3 to 5 dpf depicting the growth in diameter of a Mauthner axon (magenta−Tg(hspGFF62A:Gal4); Tg(UAS:mem-Scarlet)) and growth of myelin in thickness (green−Tg(mbp:eGFP-CAAX)) at somite 15. **B** Schematic representation of how the g-ratio for an axon is calculated. **C**–**E** Quantification of axon diameter (**C**), myelin thickness (**D**) and g-ratio (**E**) for control Mauthner axons followed from 3 to 5 dpf (RM one-way ANOVA with Tukey's multiple comparison test, $n = 18$ axons from individual animals with repeated measures at each time point. $p < 0.0001$ for (**C**) (3 dpf vs. 4 dpf and 4 dpf vs. 5 dpf), $p < 0.0001$ for (**D**) (3 dpf vs. 4 dpf and 4 dpf vs. 5 dpf), and $p = 0.0394$ for (**E**) (3 dpf vs. 5 dpf). **F** Quantification of the percentage of the Mauthner axon myelinated at 5 dpf (two-tailed unpaired $t$-test, $n = 3$ axons from individual animals for each genotype, $p = 0.9239$). **G** Representative images of the Mauthner axon and its myelin in control and neuron-specific *ipo13b* mutants at 5 dpf labelled using the transgenic lines Tg(hspGFF62A:Gal4); Tg(UAS:mem-Scarlet); Tg(mbp:eGFP-CAAX). **H**–**J** Quantification of axon diameter (**H**), myelin thickness (**I**) and g-ratio (**J**) for control and neuron-specific *ipo13b* mutant Mauthner axons at 5 dpf at somite 15 (two-tailed unpaired $t$-tests, $n = 17$ control and 18 neuron-specific mutant axons from individual animals, $p < 0.0001$ (**H**), $p = 0.7333$ (**I**), and $p < 0.0001$ (**J**)). **K** Representative electron micrographs of the Mauthner axon in control and neuron-specific *ipo13b* mutant Mauthner axons at 5 dpf at somite 15–16. **L**–**N** Quantification from electron micrographs of axon diameter (**L**), myelin thickness (**M**) and g-ratio (**N**) for control and neuron-specific *ipo13b* mutant Mauthner axons at 5 dpf at somite 15–16 (two-tailed unpaired $t$-tests, $n = 8$ control and 8 neuron-specific mutant animals, $p < 0.0001$ (**L**), $p = 0.7167$ (**M**), and $p = 0.0093$ (**N**)). All data are presented as mean values ± SD. *$p < 0.05$, **$p < 0.01$, ****$p < 0.0001$, ns = not significant. Scale bars: 10 μm (**A**, **G**), 1 μm (**K**). Source data are provided as a Source Data file.

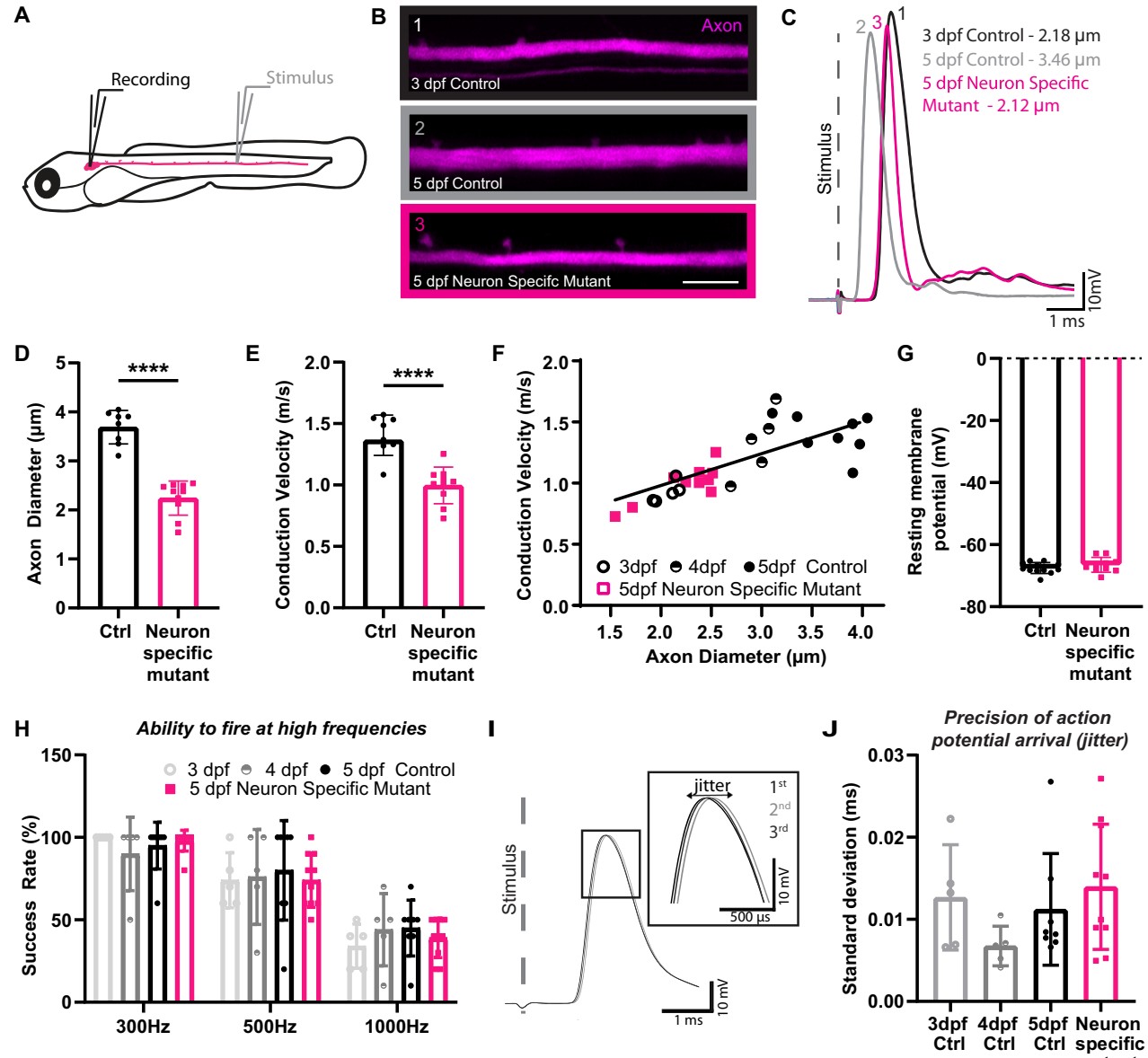

**Fig. 8 | Diameter growth drives changes to conduction speeds along myelinated axons over time. A** Schematic overview of the electrophysiological set-up for measuring conduction velocity along the Mauthner axon. **B** Representative super-resolution confocal images of Mauthner axons labelled using the transgenic line Tg(hspGFF62A:Gal4); Tg(UAS:mRFP) alongside whole-cell patch clamp traces showing action potentials generated in response to extracellular stimulation of the same axons (**C**). **D** Axon diameter measurements for Mauthner axons at 5 dpf at somite 15 in neuron-specific *ipo13b* mutant animals and control siblings (two-tailed unpaired t-test with Welch's correction, *n* = 8 control and 10 neuron-specific mutant axons from individual animals, *p* < 0.0001). **E** Conduction velocity measurements for the same axons as in (**D**) (two-tailed unpaired *t*-test with Welch's correction, *n* = 8 control and 10 neuron-specific mutant axons from individual animals, *p* < 0.0001). **F** Conduction velocity along Mauthner axon plotted against axon diameter for control axons (3–5 dpf, *n* = 18 animals) and neuron-specific *ipo13b* mutant axons (5 dpf, *n* = 10 animals). Control points are fitted with a linear regression line (R = 0.4959). Neuron-specific *ipo13b* mutants are not significantly different from controls by simple linear regression test (slope: p = 0.5931,

intercepts: p = 0.6215). **G** Resting membrane potential in control and neuron-specific *ipo13b* mutant Mauthner neurons at 5 dpf (two-tailed unpaired *t*-test, *n* = 12 control and 10 neuron-specific *ipo13b* mutant Mauthner cells from individual animals, *p* = 0.2581). **H** The success rate of action potential firing by the Mauthner neuron in response to 10 stimulations at 300 Hz, 500 Hz and 1000 Hz (two-way RM ANOVA with Tukey's multiple comparisons tests, *n* = 5 control 3 dpf, 5 control 4 dpf, 8 control 5 dpf, 10 neuron-specific mutant 5 dpf axons from individual animals with repeated measures at each frequency, no significant differences between the controls of different ages, or between any of the controls and neuron-specific *ipo13b* mutants, *p* = 0.9667). **I** Depiction of three consecutive action potentials, which have slight variations in their latency of arrival. This variation is referred to as jitter. **J** The precision of action potential arrival (jitter) along Mauthner axon (One-way ANOVA with Tukey's multiple comparisons test, *n* = 5 control 3 dpf, 5 control 4 dpf, 8 control 5dpf, 10 neuron-specific mutant 5 dpf axons from individual animals, *p* = 0.2699). Scale bars = 10 μm. All data are presented as mean values ± SD. ****p* < 0.0001. Source data are provided as a Source Data file.

delivery on ion channels to the axonal membrane) are co-regulated by importin 13b in association with its influence on axon diameter. In addition, importin 13b might independently regulate axon diameter and other features that influence conduction. For example, in rodents, neurofilaments have been shown to influence conduction by affecting

diameter and also via diameter-independent modulation of ion channels[59,81]. However, our finding that the ability to fire precise and high-frequency action potentials is not affected in neuron-specific importin 13b mutants suggests that the mechanisms that regulate the formation and maintenance of other key features of myelinated axons

that influence conduction are at least partially independent of the mechanisms regulating axon diameter growth. The tight link between diameter growth and maturation of conduction velocity observed for a myelinated axon underscores the need to better understand the regulation of axon diameter as a mechanism to control the timing of action potential propagation within neuronal circuits.

Going forward, a major challenge for the field will be to move towards a holistic view of myelinated axon structure and function, which will require an understanding of the regulation of each aspect of the entire myelinated axon unit (e.g. axon diameter, myelin, nodes of Ranvier, ion channels) individually and in relation to one another, and the specific aspects of conduction that they influence. In recent years a great deal of attention has been placed on the concept that myelinated axons and their conduction properties may be subject to ongoing modulation, including by neuronal activity. For example, a major focus has been placed on the fact that myelination by oligodendrocytes responds to neuronal activity and that this may, in turn, dynamically modulate neural circuit function; for recent reviews see[14,82–84]. However, recent studies have shown that axon diameter can also be influenced by neural activity[18,19]. Indeed, it could be argued that the dynamic regulation of axon diameter provides a much simpler means to set and adjust the conduction properties of myelinated axons, compared, for example, to regulating numerous independent myelin sheaths, or axonal sub-domains, along the entire length of a myelinated axon. In the future, it will be important to determine how each of the function-regulating aspects of myelinated axons can change over time, which aspects co-vary, which can be independently regulated, and whether different types of axons modulate their conduction properties in different ways. Zebrafish are a powerful model with which to undertake such integrated analyses.

## Methods
### Zebrafish husbandry and transgenic/mutant lines
Adult zebrafish (*Danio rerio*) were housed and maintained in accordance with standard procedures in the Queen's Medical Research Institute zebrafish facility, University of Edinburgh. All experiments were performed in compliance with the UK Home Office, according to its regulations under project licences 60/4035, 70/8436 and PP5258250. Adult zebrafish were subject to a 14/10 h light/dark cycle. Embryos were produced by pairwise matings and were staged according to days post-fertilisation (dpf). Embryos were raised at 28.5°C, with 50 embryos (or less) per 10 cm petri dish in ~45 mL of 10 mM HEPES-buffered E3 embryo medium or conditioned aquarium water with methylene blue. All experiments used zebrafish larvae between 2 and 7 dpf on a wild type (AB/WIK/TL) or nacre$^{-/-}$ [85] background. At these ages, sexual differentiation of zebrafish has not yet occurred.

Throughout the text and in figures, 'Tg' denotes a stable, germline-inserted transgenic line. The following lines were used in this study: Tg(mbp:eGFP-CAAX)[44], Tg(hspGFF62A:Gal4)[86,87], Tg(UAS:mRFP)[86], Tg(UAS:GFP)[86], Tg(UAS:mem-Scarlet) (generated in this study). The *ipo13b*$^{ue57}$ mutant was identified during an ENU-based forward genetic screen, described below. The *ipo13b*$^{ue76}$ and *ipo13b*$^{ue77}$ mutants were generated using CRISPR/Cas9-based gene targeting, as described below. In experiments using *ipo13b*$^{ue57}$, *ipo13b*$^{ue76}$, or *ipo13b*$^{ue77}$ animals, "control" refers to a mixture of wild-type and heterozygous siblings.

### ENU mutagenesis screen
The ENU mutagenesis screen in which the *ipo13b*$^{ue57}$ mutant was generated was previously published[46,47]. Briefly, 10 adult AB males were mutagenized with 3.5 mM ENU for 1 h per week over three consecutive weeks. Assessment of mutagenesis efficiency was made by crossing with carriers of a pigment mutation, *sox10*$^{cls}$[88], and well-mutagenised males were crossed with AB females to generate the F1 generation. F1

individuals were bred with Tg(mbp:eGFP-CAAX) animals to introduce a myelin reporter into the mutagenized stocks and generate individual F2 families. In total, 212 F2 families were generated, from which 946 clutches were screened at 5 dpf for disruption to myelinated axons. For extensive protocol details on zebrafish ENU-based mutagenesis forward genetic screens, see[45].

### *ue57* mapping-by-sequencing
Following an outcross to WIK, pooled DNA from mutant recombinants and non-mutant recombinants was sequenced separately on an Illumina HiSeq4000 (Edinburgh Genomics). We processed this data through a modified version of the Variant Discovery Mapping Cloud-Map pipeline[89] on an in-house Galaxy server using the Zv9/danRer7 genome and annotation. For both the Variant Discovery Mapping plots and assessing the list of candidate variants, we subtracted a list of wild-type variants compiled from sequencing of the EKW strain plus previously published data[90–92]. The mapping identified a 5 Mb mapping region on Chr 20 between 45 and 50 Mb. From the variant list within this region, we filtered for prospective missense and nonsense mutations likely to result in a strong loss of function of encoded proteins. This candidate list was further filtered by excluding polymorphisms found in other mutants that we sequenced that derived from the same ENU screen. This revealed seven protein-coding variants within the region, of which only the variant in *ipo13b* resulted in a predicted nonsense mutation.

### *ue57* genotyping
*ue57* fish were genotyped as follows. A 205 bp PCR product including the *ue57* mutation, was amplified using the following primers: Fwd 5′-CCACTAATAAAGACTATGTTTTCTCCTTTTT**G**CTC and Rev 5′-CTTCAGTTTCCGACACAGTATTG. Note the bolded G in the fwd primer represents of T > G mismatch introduced to generate an MwoI restriction enzyme cut site in the wild-type DNA sequence. 10 μM of each primer was combined with OneTaq QuickLoad Master Mix (NEB #M0486) and run in a thermocycler as follows: 94°C for 2 min; 36 cycles of 94°C for 30 s, 51°C for 20 s, 68°C for 10 s; and 68°C for 5 min. The PCR product was digested with 1 unit of MwoI (NEB #R0573S) for 2 h at 60°C, then run on a 3% agarose gel at 150 V for 45 min. The *ue57* mutation disrupts the MwoI cut site resulting in a single DNA band of ~205 bp, while wild type fish have two DNA bands of 174 bp and 31 bp.

### CRISPR/Cas9 based targeting
To independently disrupt *ipo13b* function, we designed a sgRNA (GTTGCTGACGGTTCTTCCAG) targeting a region in exon 3 previously shown to interact with ran-GTP[57], the binding to which is essential for importin 13 function. The sgRNAs were generated using the MEGAshortscript T7 Transcription Kit (Invitrogen #AM1354) according to manufacturer's instructions and purified using MEGAclear Transcription Clean-up Kit (Invitrogen #AM1908) according to manufacturer's instructions. To synthesise the template DNA required for the sgRNA in vitro transcription, a two-oligo PCR method was used, as described in[93]. The *ipo13b*-specific oligo sequence was 5′- AATTAATACGACTCACTATA<u>GGTGCTGACGGTTCTTCCAG</u>GTTTTAGAGCTAGAAATAGC. The scaffold oligo sequence was 5′-GATCCGCACCGACTCGGTGCCACTTTTTCAAGTTGATAACGGACTAGCCTTATTTTAACTTGCTATTTCTAGCTCTAAAAC. The PCR reaction was performed using Phusion High-Fidelity DNA Polymerase (NEB #M0530) with 10 μM of each primer (synthesised by IDT) and run in a thermocycler as follows: 95°C for 2 min; 30 cycles of 95°C for 10 s, 60°C for 20 s, 72°C for 10 s; and 72°C for 5 min. The full-length PCR product was purified using Monarch Gel Extraction Kit (NEB #T1020), according to the manufacturer's instructions.

In total, 500 ng of sgRNA was combined with 20 μM EnGen Cas9 NLS (NEB #M0646) in 1× Cas9 Nuclease Reaction Buffer (NEB

#B0386A) (total volume 3 µl) and incubated at 37 °C for 5 min, then transferred to ice. Approximately 1 nL of the sgRNA-Cas9 ribonucleoprotein complex was injected into Tg(mbp:eGFP-CAAX) embryos at the one-cell stage. Embryos were raised to adulthood and outcrossed to nacre⁻/⁻ fish to generate the F1 generation. F1 fish were screened for the presence of CRISPR/Cas9 induced mutations by amplifying a region of exon 3 using the primers 5′-GACTCCCTCAAATCCCAGCTC and 5′-GACAGACACTTGAGCACACG. 10 µM of primers were combined with OneTaq QuickLoad Master Mix (NEB #M0486) and run in a thermocycler as follows: 94 °C for 2 min; 30 cycles of 94 °C for 30 s, 57 °C for 30 s, 68 °C for 30 s; and 68 °C for 5 min. The 385 bp PCR product was digested with 1 unit of Hpy188III (NEB #R0622) for 2 h at 37 °C, then run on a 2% agarose gel at 180 V for 45 min. Mutations disrupted a Hpy188III cut site resulting in a DNA band of ~231 bp, rather than two DNA bands of 162 bp and 69 bp. The exact indel size and location were determined by Sanger sequencing of the 385 bp PCR product followed by analysis using CrispID (crisped.gbiomed.kuleuven.be)[94] to de-convolute the overlapping spectra from the wild type and mutant alleles. Two different mutations (*ipo13b*$^{ue76}$, c.486delA and *ipo13b*$^{ue77}$, c.477_490del) were selected to propagate F2 generations, which were then in-crossed to generate the mutants used in this study. Both mutations result in frameshifts and premature stop codons.

## Generation of UAS:mem-Scarlet transgenic line

The UAS:mem-Scarlet expression vector was generated using Gateway cloning. mScarlet was PCR amplified from the plasmid pmScarlet_C1 (Addgene plasmid #85042) while also adding a fyn myristolyation sequence (a membrane targeting sequence) to the 5′ end using the following primers: mem-Scarlet_F 5′-GGGGACAAGTTTGTA-CAAAAAAGCAGGCTGCCACCATGGGCTGTGTGCAATGTAAGGATAAA-GAAGCAACAAAACTGACGGTGAGCAAGGGCGAGGCAG 3′ and mem-Scarlet_R 5′-GGGGACCACTTTGTACAAGAAAGCTGGGTTTACTTGTA-CAGCTCGTCCATGCCG. A middle entry vector (pME_mem-Scarlet) was generated using 150 ng of purified PCR product with 150 ng of pDONR221 in a BP reaction using BP clonase II performed overnight at room temperature. To generate the final Tol2 expression vector (UAS:mem-Scarlet), 20 fmol of each of the following entry and destination vectors were combined in a LR reaction using LR Clonase II Plus performed overnight at room temperature: p5E_10UAS (Tol2Kit #327)[95], pME_mem-Scarlet, p3E-polyA (Tol2Kit #302), and pDest-Tol2pA2-cryaa:mCherry (Addgene #64023). Note the destination vector includes the red eye marker cryaa:mCherry for screening of transgenics. Clones were tested for correct recombination by digestion with restriction enzymes. The transgenic line was generated by injecting 10–20 pg of the final UAS:mem-Scarlet vector with 50 pg *tol2* transposase into nacre⁻/⁻ zebrafish eggs at the one-cell stage. Founder animals were identified by outcrossing and screening for the red eye marker, then crossed with several different in-house Gal4 transgenic lines to confirm mem-Scarlet expression to select the best founder. Experiments were performed with the F2 and F3 generations. Multiple copies of the transgenic insertion were present in these generations.

## Live-imaging

Larvae were anaesthetised with 600 µM tricaine in E3 embryo medium and immobilised in 1.3–1.5% low melting-point agarose on a glass coverslip, which was suspended over a microscope slide using high vacuum silicone grease to create a well containing E3 embryo medium and 600 µM tricaine. Z-stacks (with optimal z-step) were obtained using a Zeiss LSM880 microscope with Airyscan FAST in super-resolution mode, using a 20× objective lens (Zeiss Plan-Apochromat 20× dry, NA = 0.8), and processed using the default Airyscan processing settings (Zen Black 2.3, Zeiss). All Mauthner, MiM1, and Mid3i axon images were taken from a lateral view of the spinal cord centred around somite 15. Mauthner, MiM1, and Mid3i cell bodies were imaged from the dorsal surface of the hindbrain. All lateral view images depict

the anterior on the left and dorsal of the top, while dorsal view images depict the anterior on the top. Figure panels were prepared using Fiji (v1.51n) and Adobe Illustrator 2020 (24.0.2).

## Quantification of axon diameter

Axon diameter from AiryScan FAST confocal images were measured using the following scripts (available at https://github.com/jasonjearly/Axon_Caliber/releases/tag/v1.0.0[96]), which were written in ImageJ Macro Language using the open source image analysis software Fiji (v1.51n)[97,98].

A "Split Axons Tool" was produced to allow the splitting of two adjacent Mauthner (or MiM1/Mid3i axons) in whole larvae z-stack datasets. To achieve this, the macro rotates the dataset to give the top-down (x–z plane) view, performs a maximum intensity projection, presents this to the user and asks for a point to be placed between the two axons. The macro then samples the image repeatedly along the z-axis in user-configurable x-widths to identify local minima between peak fluorescence. By doing so, the fluorescence trough between axons is mapped. The minima closest to the user-selected point are selected and used to create two separate selections on duplicates of the original x–z rotated z-stack for the upper and lower Mauthner axons, where the pixels outside the selections for the respective axons are set to zero grey values, before rotating these new z-stacks back to the front-facing x–y plane for saving and analysis.

In order to measure axon diameter, two separate ImageJ tools were generated. Firstly, the "Axon Trace Tool" was written to trace the approximate centre point of the axon along its length. To do this, the edges of the axon are identified by forming an intensity profile along the y-axis profile using the Plot Profile function in ImageJ (with a user-configurable thickness in the x-axis) and the centre point for this location on the x-axis, given by the midpoint of the axon edges. This is then repeated along the x-axis to give a trace of the centre point of the axon for any given x-coordinate.

Secondly, the "Axon Calibre Tool" was written to measure the average axon diameter along the length of the selected axon and was based on the method used to measure axon diameter in[18]. This macro takes the axon trace produced by the Axon Trace Tool and, for each location along the x-axis, calculates the trajectory of the axon based on a user-configurable number of x–y centre coordinates before and after the current location. The macro then plots a line (of user-configurable width) perpendicular to the calculated trajectory and extracts the fluorescence profile along this line. The edges of the axon are then extracted from this profile, giving the width at this location along the x-axis. The macro then repeats this process along the x-axis and provides the diameter at each x–y coordinate, as well as the average thickness of the axon. Unless otherwise indicated, axon diameter was measured for only one axon per fish, selecting the axon located closest to the imaging objective.

## *ue57* rescue experiments

To rescue the *ue57* phenotype, 125 pg of *ipo13b* mRNA was injected into the fertilised single cell of eggs from Tg(mbp:eGFP-CAAX) fish. At 5 dpf, fish were imaged at somite 15 and the myelinated axon diameter was calculated by tracing the area of the myelinated Mauthner axon and dividing by its length.

## Quantification of axon length and percentage of axon myelinated

Overlapping images of the entire Mauthner neuron were obtained in laterally mounted fish, as outlined above in "Live-Imaging". The "Split Axon" tool (described above) or 3D segmentation was used to isolate the axon of interest[99], and images were stitched from the file positions using the Grid/Collection Stitching plugin Fiji (v1.51n). The axon was traced to obtain a length measurement. To calculate the percentage of the axon that was myelinated, unmyelinated and myelinated regions of

the axon were measured using the Tg(mbp:eGFP-CAAX) reporter line to label myelin.

## Quantification of cell body size

The same Mauthner neuron was imaged from the dorsal surface of the zebrafish hindbrain at 3, 4, and 5 dpf. A maximum intensity projection of the z-stack was generated, and the axon cropped out of the image to ensure the individual performing the analysis remained blinded to the genotypes. The cell body and lateral dendrite were traced, and an area measurement was obtained using Fiji (v1.51n).

To quantify cell body volume, cell bodies were segmented manually, using the manual brush tool of Arivis Vision4D, slice by slice in the z-stack image. Clear, in-focus, cell body fluorescence was manually masked using the brush in sequential z-slices and overlapping masks were reconstructed into 3D objects for volume measurement.

## Quantification of g-ratios

G-ratios were measured from longitudinal images of the Mauthner axon imaged using a Zeiss LSM880 microscope with Airyscan FAST in super-resolution mode, using a 20× objective lens and 1.8× zoom. Analysis was performed in Fiji (v1.51n). Z-stacks were converted to maximum intensity projections. The area of the axon over a 50–70 μm stretch of the axon (membrane labelled using Tg(hspGFF62A:Gal4); Tg(UAS:mem-Scarlet)) was traced and measured using the polygon selection tool, and the average diameter determined by dividing the area by the axon length. The same was done to find the average diameter of the myelin + Mauthner axon, which was labelled using Tg(mbp:eGFP-CAAX). For time-course analysis the exact same region of the axon was analysed at each time point, using the synaptic boutons which protrude from the axon as landmarks to identify the same region. G-ratios were determined by dividing the diameter of the axon by the diameter of the myelin + axon. Myelin thickness was determined by subtracting the diameter of the axon from the diameter of the myelin + axon and dividing by two (also see Supplemental Fig. 1).

As a second approach, g-ratios for the Mauthner axon were also measured in electron microscopy images. Axon diameter was measured as outlined below in "Transmission Electron Microscopy". Myelin thickness was measured in 3–4 regions without fixation artefacts and averaged. G-ratio was solved by dividing the diameter of the axon by the diameter of the myelin + axon (diameter myelin + axon = 2*myelin thickness + axon diameter) (also see Supplemental Fig. 1).

## Transmission electron microscopy

Zebrafish tissue was prepared for transmission electron microscopy according to a previously published protocol[100]. Briefly, zebrafish embryos were terminally anaesthetised in tricaine, heads removed for genotyping, and trunks incubated with microwave stimulation in a primary fixative of 4% paraformaldehyde + 2% glutaraldehyde in 0.1 M sodium cacodylate buffer. Tissue was then stored for up to 3 weeks in primary fixative at 4 °C. Samples were washed in 0.1 M Cacodylate buffer and incubated with microwave stimulation in secondary fixative of 2% osmium tetroxide in 0.1 M sodium cacodylate/0.1 M imidazole buffer pH7.5, then left 3 h at room temperature. Samples were washed with distilled water, then stained en bloc with a saturated (8%) uranyl acetate solution with microwave stimulation followed by overnight incubation at room temperature. Next, samples were dehydrated with an ethanol series and acetone using microwave stimulation. Samples were embedded in Embed-812 resin (Electron Microscopy Sciences), and silver sections (70–80 nm in thickness) were cut using either a Reichert Jung or Leica Ultracut Microtome and a Diatome diamond knife. Sections were mounted on hexagonal copper electron microscopy grids (200 Mesh Grids, Agar Scientific). Mounted sections were stained in uranyl acetate and Sato's lead stain. TEM imaging was performed at the University of Edinburgh Biology Scanning Electron Microscope Facility using a Jeol JEM1400 Plus Transmission Electron Microscope. The dorsal and ventral spinal cord were imaged at 12000x magnification and the Mauthner axon at 15000x magnification. In order to create panoramic views, individual electron micrograph tiles were aligned using the automated photomerge tool in Adobe Photoshop 2020 (21.0.2). To assess axon diameter, axonal areas were measured by tracing axons in Fiji (v1.51n), which were then used to calculate diameter using the equation $diameter = 2*\sqrt{(\frac{Area}{\pi})}$.

## Neurofilament analysis

ROIs of 430 pixels x 430 pixels which were absent of mitochondria or other organelles, were selected from each electron micrograph image of the Mauthner axon for quantification. Neurofilaments in electron micrographs were detected using the ML-segmenter algorithm in Arivis Vision4D. To train the model in the algorithm, neurofilament and non-neurofilament class examples were annotated using a brush tool across 5 images. The trained model was used to segment neurofilament objects across the entire ROI image batch. Coordinates of neurofilament objects were exported and used to calculate the nearest neighbours using the built-in Nearest[] function in Wolfram Mathematica 13.0. Neurofilament density was calculated using the number of neurofilaments within the ROI.

## Cell-type specific targeting of *ipo13b* in neurons

Cas9 and sgRNA expression vectors were generated using a previously described strategy and protocol[101,102]. Firstly, a U6-based sgRNA construct expressing 3 sgRNAs targeting *ipo13b* was generated using Golden gate cloning. The following sgRNA sequences were used, which were designed using the CRISPRscan tool[103] and selected based on low prediction for off-targeting: *ipo13b* sgRNA1: GGTGCCTGAGGCCTGGCCGG (exon 3), *ipo13b* sgRNA2: GGTGCTGACGGTTCTTCCAG (exon 3), *ipo13b* sgRNA3: GGTTGAGTCGAGGACTACAG (exon 21). Each sgRNA was tested for efficiency prior to cloning by using them to generate F0 generation mutants (crispants), which phenocopied the *ipo13b*$^{ue57}$ mutant phenotype. To clone the sgRNA sequences into the U6 expression vectors, annealed oligos containing the sgRNA and BsmBI sequences were generated using the following primer pairs: 5′-TTCGGTGCCTGAGGCC TGGCCGG & 5′-AAACCCGGCCAGGCCTCAGGCAC, 5′-TTCGGTGCTGAC GGTTCTTCCAG & 5′-AAACCTGGAAGAACCGTCAGCAC, 5′-TTCGGGA GTCAAAGACATTGTGA & 5′-AAACTCACAATGTCTTTGACTCC. 200 μM of each oligo were mixed together in a 20 μl reaction with NEB Buffer 2.1 and incubated at 95 °C for 5 min, then decreased to 50 °C at 0.1 °C/s, followed by a 10 min incubation at 50 °C. The annealed oligos were then ligated into U6 promotor-based cassettes: *ipo13b* sgRNA1 into pU6a:sgRNA#1 (Addgene #64245), *ipo13b* sgRNA2 into pU6a:sgRNA#2 (Addgene #64246), and *ipo13b* sgRNA3 into pU6b:sgRNA#3 (Addgene #64247). Ligation reactions were carried out by mixing 1 μL 10× NEB CutSmart buffer, 1 μL T4 DNA ligase buffer, 100 ng U6 plasmid, 1 μl annealed oligos, 0.3 μL T4 DNA ligase, 0.3 μL BsmBI, 0.2 μl PstI, and 0.2 μL SalI in a 10 μL reaction, then incubating for 3 cycles of 37 °C for 20 min and 16 °C for 15 min. This was followed by 37 °C for 10 min, 55 °C for 15 min and 80 °C for 15 min.

To construct the final sgRNA expressing vector containing all 3 sgRNAs, 50 ng of pGGDestTol2LC-3sgRNA (Addgene #64241) was combined with 100 ng of each pU6:sgRNA vector generated above, 2 μL 10× NEB CutSmart Buffer, 2 μL T4 DNA ligase buffer, 1 μL T4 DNA ligase, and 1 μL BsaI in a 20 μL reaction. This was incubated with 3 cycles of 37 °C for 20 min and 16 °C for 15 min, followed by 15 min at 80 °C. The destination vector of pGGDestTol2LC-3sgRNA contains a cerulean eye marker (cryaa:cerulean) for screening transgenics. The sequence of the final expression vector was confirmed by sequencing before use.

To generate the construct expressing Cas9 in neurons, the Tol2-based multisite Gateway system was used. 20 fmol of each of the following entry and destination vectors were combined in a LR reaction using LR Clonase II Plus performed overnight at room temperature: p5E_nbt (*Xenopus* neural-specific beta tubulin, also referred to as

XIa.Tubb[104,105]), pME_cas9 (Addgene #64237), p3E-polyA (Tol2Kit #302), and pDestTol2pA2-cryaa:mCherry (Addgene #64023). Note, that the destination vector includes the red eye marker cryaa:mCherry for screening of transgenics. Clones were tested for correct recombination by digestion with restriction enzymes.

Two transgenic lines, Tg(U6:3sgRNA-ipo13b) and Tg(nbt:cas9), were generated by injecting 10–20 pg of the expression vector with 50 pg *tol2* transposase into Tg(mbp:eGFP-CAAX) zebrafish eggs at the one-cell stage. Founder animals were identified by outcrossing and screening for the red or blue eye markers. Founders were outcrossed to Tg(hspGFF62A:Gal4); Tg(UAS:mRFP) to generate F1 offspring that carried either Tg(U6:3sgRNA-ipo13b) or Tg(nbt:cas9), as well as Tg(hspGFF62A:Gal4); Tg(UAS:mRFP) and Tg(mbp:eGFP-CAAX) to label the Mauthner axon and myelin. Neuron-specific mutants were generated by crossing Tg(U6:3sgRNA-ipo13b) and Tg(nbt:cas9) lines and screening for the presence of both red and cerulean eye markers. All experiments were performed using parents from the F1 to F3 generations. Multiple insertions were present in these generations. Throughout the paper "neuron-specific ipo13b mutants" refers to fish positive for both the (nbt:cas9) and (U6:3sgRNA-ipo13b) transgenes, while "control" refers to siblings expressing only the (nbt:cas9) or (U6:3sgRNA-ipo13b) transgene alone.

### Electrophysiology

Zebrafish were dissected as described previously[106] to access the Mauthner neuron. For experiments using neuron-specific *ipo13b* mutants, fish were pre-screened and only those exhibiting reduced axon diameter were used for electrophysiology. In short, 2–5 dpf anaesthetised zebrafish were laid on their sides on a Sylgard dish and secured using tungsten pins through their notochords in a dissection solution containing the following: 134 mM NaCl, 2.9 KCl mM, 2.1 CaCl$_2$ mM, 1.2 MgCl$_2$ mM, 10 mM HEPES, 10 mM glucose and 600 μM tricaine, adjusted to pH 7.8 with NaOH. Their eyes and lower and upper jaws were removed using forceps to expose the ventral surface of the hindbrain, which was secured with an additional tungsten pin. The Mauthner axons were exposed, alongside motor neurons, as described previously[107]. A dissecting tungsten pin was used to remove the skin and the muscle overlaying the Mauthner axons and motoneurons in a single segment. Following the dissection, zebrafish together with their recording chamber were moved to the recording rig and washed with extracellular solution containing the following: 134 mM NaCl, 2.9 mM KCl, 2.1 mM CaCl$_2$, 1.2 mM MgCl$_2$, 10 mM HEPES, 10 mM glucose and 15 μM tubocurarine. The cells were visualised using an Olympus microscope capable of Differential Interference Contrast (DIC) using a 60× water immersion, NA = 1 objective lens and a ROLERA Bolt sCMOS camera (QImaging) with Q capture software. The stimulating electrode filled with an extracellular solution was then positioned in the spinal cord, touching the exposed neurons underneath. Mauthner whole-cell recordings were performed with thick-walled borosilicate glass pipettes pulled to 6–10 MΩ. The internal solution contained the following: 125 mM K-gluconate, 15 mM KCl, 10 mM HEPES, 5 mM EGTA, 2 mM MgCl$_2$, 0.4 mM NaGTP, 2 mM NaATP, and 10 mM Na-phosphocreatine, adjusted to pH 7.4 with KOH. A 270 s recording was performed in a current clamp configuration, and the cell resting membrane potential was established as an average of the first 10 s of the recording if the cell did not fire during that time. To measure the conduction velocity along the Mauthner axon, 270 s after breaking through the cell, the zebrafish were washed with recording solution containing the following: 134 mM NaCl, 2.9 mM KCl, 2.1 mM CaCl$_2$, 1.2 mM MgCl$_2$, 10 mM HEPES, 10 mM Glucose. Blockers of fast synaptic transmission were supplemented to this solution: 50 μM AP5, 20 μM strychnine, 100 μM picrotoxin and 50 μM CNQX. The antidromic Mauthner action potentials were recorded following the field stimulation by the stimulating electrode connected to DS2A Isolated Voltage Stimulator (Digitimer) in the spinal cord. The action potentials were recoded using Clampex 10.6 (Molecular Devices) at a 100 kHz sampling rate and filtered at 2 kHz using a MultiClamp 700B amplifier (Molecular Devices). 30 consecutive action potentials were recoded every 5 s using Clampex 10.6 (Molecular Devices) at a 100 kHz sampling rate and filtered at 2 kHz using a MultiClamp 700B amplifier (Molecular Devices). At the end of the recording, images of the zebrafish were obtained with a 4X objective and stitched using Adobe Photoshop. The resulting image was then transferred to Fiji, and the distance between the stimulating and recording electrode was measured. The latency of the action potential peak was calculated by the in-house built MATLAB Script[70] available at https://github.com/skotuke/Mauthner_analysis/releases/tag/v1.0.0[108]. The conduction velocity of action potentials was calculated by dividing the distance between the electrodes by the latency from the action potential peak to the stimulus artefact. The conduction velocity of the first action potential recorded from the cell was included in the analysis if the cell fired at least 15 action potentials following 30 extracellular stimulations. Only one cell per fish was recorded and included in the electrophysiological analysis. For the analysis of action potential fidelity, consecutive trains of 10 stimuli were delivered at 1, 10, 100, 300, 500 and 1000 Hz every 30 s. Recordings were made at a 20 kHz sampling rate and filtered at 2 kHz. The number of action potentials was calculated using Clampfit software, and the action potential success rate was calculated as the number of action potentials fired out of 10. The precision of action potential arrival was calculated as the standard deviation of 30 consecutive action potential latencies from the stimulus artefact.

### Behavioural analysis

At 4 dpf individual larvae were added to each well of a 24-well plate in 1.5 mL of 10 mM HEPES-buffered E3 embryo medium. The next day, a 20 min period of free swimming was recorded using a DanioVision Box and Ethovision XT 14 software (Noldus Information Technology, Netherlands). Ethovision XT 14 software was used to determine the percentage of time spent moving, the number of swim bouts initiated, and swim bout durations. A track smoothing profile was added, setting the minimal distance moved to >0.20 mm, and the maximum distance moved to >9.20 mm. Movement was calculated using an averaging interval of 3 samples, start velocity of 5.00 mm/s and stop velocity of 1.00 mm/s. Only fish with inflated swim bladders were analyzed.

### Statistical analysis

Statistical analysis was performed using GraphPad Prism 8 or Graph-Pad Prism 10 software (up to version 10.1.2), with results included in the figure legends and Source Data file. Unless otherwise indicated, n number represents values from independent fish. Only one axon was analysed per animal, with the exception of the electron microscopy datasets in which an average value for both Mauthner axons was used. All experiments were performed on animals from at least two different clutches of fish. All data are presented as mean ± standard deviation, aside from behavioural data, which is presented as box and whisker plots indicating the median, 25th and 75th percentiles, max and min. *N* values, *p* values and statistical tests are noted in the figure legends. Significance was defined as $p < 0.05$.

### Reporting summary

Further information on research design is available in the Nature Portfolio Reporting Summary linked to this article.

## Data availability

Data generated in this study are provided within the Source Data file. All requests for raw images or materials should be addressed to the corresponding author. Source data are provided with this paper.

## Code availability

Custom code used in the analyses of this study is available at https://github.com/jasonjearly/Axon_Caliber/releases/tag/v1.0.0 https://doi.org/10.5281/zenodo.10570003[96] and https://github.com/skotuke/Mauthner_analysis/releases/tag/v1.0.0 https://doi.org/10.5281/zenodo.10479177[108].

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

## Acknowledgements

We thank members of the Lyons laboratory for feedback and the University of Edinburgh BVS Zebrafish Facility, Zebrafish Imaging and Screening Facility, and Transmission Electron Microscopy Facility for expert assistance. This work was supported by Wellcome Trust Senior Research Fellowships (102836/Z/13/Z and 214244/Z/18/Z) and a Lister Institute Research Prize to D.A.L. and a Wellcome Trust Research Career Development Fellowship (095722/Z/11/Z) and a Wellcome Senior Fellow in Basic Biomedical Science (207483/Z/17/Z) to R.J.P. J.M.B was supported by a Canadian Institute of Health Research a postdoctoral fellowship and a Multiple Sclerosis Society of Canada/Fonds de la recherche du Quebec-Sante postdoctoral fellowship. M.R.L. was supported by a Royal Society of Edinburgh/Caledonian Research Fund Personal Research Fellowship.

## Author contributions

J.M.B. performed the experiments, analyzed, and interpreted the data related to the various importin 13 mutants, with the exception of electrophysiology experiments which were performed by D.S. and supervised by M.R.L. S.B.-K. supported the initial characterisation of the ipo13b[ue57] mutants. S.B.-K., L.K. and M.R.-B. performed the ENU-mutagenesis screen, J.J.E. and D.S. wrote code and assisted with image analysis, R.J.P. performed mapping of the ENU mutation, D.A.L. supervised the project and obtained funding. J.M.B and D.A.L wrote the paper with editing by all other authors.

## Competing interests

The authors declare no competing interests.
