## [Peer Review File · Nature Communications]

Importin 13-dependent Axon Diameter Growth Regulates Conduction Speeds along Myelinated CNS AxonsREVIEWER COMMENTS

Reviewer #1 (Remarks to the Author):

Bin et al identify importin-13b (ipo13b) as a regulator of axon diameter and action potential conduction speed. A diverse set of experimental strategies were used to describe the impact of ipo13b loss and functional consequences on neuronal physiology. EM was used to show that ipo13b broadly affects axon diameters throughout the spinal cord. The authors also take a more focused approach using the Mauthner neuron, allowing reliable characterization of a single identifiable neuron, and perform super resolution imaging and in vivo electrophysiology to show detailed structural and functional consequences in ipo13b mutants or following neuron-specific knockdown through development, which provided complementary results. Structural analysis demonstrates that ipo13b is critical for the growth of axon diameter during development, yet not other structural features of neurons. Functional experiments, using electrophysiology, also demonstrate specific effects, whereas only conduction speed is impacted yet not the ability to entrain sustained stimulation. Collectively, this is a very nice study that implements many of the strengths of the zebrafish system, forward genetics, optical imaging, electrophysiology etc., to uncover novel mechanisms regulating fundamental properties of nervous system function. Genetic regulation of axon diameter is poorly understood and uniquely challenging to address, which makes this current study highly impactful and likely to be of broad interest. The experiments and analysis are rigorous and provide strong support to the conclusion drawn. The manuscript is appropriate for publication in Nature Communications, yet there are several points which I think would help strengthen conclusions and clarify interpretations.

1. The authors use multiple CRISPR alleles to recapitulate the screen mutant phenotype as evidence that ipo13b is the basis of the observed axon diameter phenotypes. A rescue experiment would provide strong supporting evidence. An ideal experiment would be mosaic rescue in 1 but not both Mauthner neurons. However, with the strength of the evidence already provided any form of rescue experiment would be a solid addition.

2. Ipo13b mutants show no change in Mauthner cell body area, yet volume or at least a measure from an orthogonal view would strengthen the claim that ipo13b does not impact cell body growth. Similarly, quantifying the length, or any other parameter, of the lateral dendrite would help support that ipo13b is highly specific to regulating axon diameter.

3. Some additional measures from the physiological recordings would be helpful. For example, action potential rise time and half-width.

4. Some discussion is provided regarding how importin 13b may be modulating axon diameter and the authors do a good job at highlighting how challenging this mechanism may be to resolve. However, this discussion is lean and it would be rather interesting to expand upon putative ways importin 13b may be regulating axon diameter.

Minor points:

1. In Figure 5, the changes to neurofilament density is interesting, yet the data and interpretation is a bit confusing (eg increased measures in mutants – Figure 5C, yet overall this is a decrease due to the decreased axon volume). It may be worth seeing the images Figure 5A-B with matched as well as the currently shown adjusted scales. Potentially an additional figure where the data is scaled to volume to provide a clear visual representation of the overall decrease in neurofilaments.

2. First results section the authors state “project their axons down the entire length of the contralateral spinal cord (Figure 1B)”. Figure 1B as it does not show the projection. I suggest either adding an additional image or simply add a citation for clarity.

3. Figure 2D the authors call the behavioral result “display reduced motility”, yet this is somewhat confusing. The 5 dpf larvae spend less time moving, which may be more accurate as reduced time

swimming or fewer swim initiations. Conversely, when swimming is engaged the overall motility (distance, velocity, etc) could be normal or reduced. No experiments are needed yet a text change would clarify.

4. Early in development mutant axon diameters are comparable to siblings. Are early behaviors similarly comparable?

5. Do the authors think motoneurons are impacted by the loss of ipo13b? Experiments are not needed, yet may be an interesting point of discussion and potential role of ipo13b in the peripheral nervous system.

6. In the results expanding on what jitter is and why it is an important metric would be helpful to a reader

Reviewer #2 (Remarks to the Author):

This manuscript describes the results of a mutagenesis screen in zebrafish to identify genes that influence axon diameter. Using an ENU forward genetics screen, they map one mutant displaying reduced diameter of Mauthner cell axons to the importin 13b locus. Importin 13 is a chaperone protein that assists in moving a variety of proteins, including transcription factors, from the cytoplasm to the nucleus and has been shown to play a role in neuronal development and synaptic function. Validation is solid, as two CRISPR mutants at different sites in the gene exhibit similar growth deficits, and the reduction in axon diameter was also observed in other neurons in the whole body Ipo13bue57 mutant, with the largest decline seen in larger diameter axons. The decline in axon diameter growth was accompanied by a reduction in overall myelin thickness, although this was not quantified. Ipo13 is however, widely expressed throughout the body, so complex interactions could be responsible for this phenotype. To address this, the authors show using neuronal specific CRISPR deletion that neuron-specific loss accounts for the most of the change in axon diameter.

Using cell attached patch recordings of back propagating action potentials from Mauthner cells, the authors find Ipo13bue57 mutants exhibit markedly slower conduction velocity. Control and mutant fish at a stage when the axons have a similar diameter had similar conduction velocities, suggesting that it is likely the change in axon diameter, rather than other possible aspects of Ipo13b function that are responsible for determining the speed of AP propagation.

Together, these studies introduce fluorescent labeling of Mauthner cells as a means to explore the molecular mechanisms that control axon diameter and identify importin13b as a key upstream regulatory protein in this process. Further, the careful physiological measurements provide evidence that axon diameter is a major determining factor in controlling the conduction velocity of action potentials.

The text is written with exceptional clarity and brevity, the results were analyzed rigorously, and the figures well illustrated. I have only a few suggestions for improvement.

Major

1. Given the importance of myelination on conduction velocity, it is surprising that the studies do not provide quantitative measures of myelination that coincide with the axon diameter changes. Ideally, one would like to see Mauthner cell g-ratio plots across dpf, although it does not appear from the electron microscopy images provided that myelin was sufficiently persevered. I am not an expert in zebrafish EM, so maybe this type of analysis isn't possible. If not, a light level estimation of myelin thickness across age and genotype would be helpful.

2. The description and analysis of the neuron specific Ipo13b mutants is a bit stunted. It The authors suggest that differences between complete and neuron specific deletions "reflect the timing or extent of loss of importin 13b function, or additional non-cell autonomous contributions of importin 13b." It would be helpful to include quantifications of axon diameter in the text, so that

the reader doesn't have to estimate from the figure (in general, this would be helpful for many of the experiments, as there few statistics or statistical tests included in the text). A limitation here is that there are no data to assess various possibilities, so one is left wondering what accounts for the difference between complete and neuron specific deletions. Immunocytochemistry against Importin 13b would be beneficial to the study and would help address these possibilities.

3. One of the main conclusions of the study is that the increases in conduction velocity along the myelinated Mauthner axon are primarily driven by its growth in diameter. This is based on the observation that control and knockout axons of the same diameter exhibited similar conduction velocities. I feel that this conclusion oversimplifies considerably, as it is possible that importin13b controls the expression of many genes that together coordinate axonal maturation. If genes encoding ion channels, transporters, synaptic release (target derived growth factor sensitivity), etc. are co-regulated by importin13b (for example, by a key transcription factor), then their expression could also scale with diameter. The implication from the text is that axon diameter is driving these changes in functional properties, but there is limited evidence in the study to test this hypothesis. It seems more likely that many aspects of axonal maturation are co-regulated by Ipo13b. This possibility should be addressed more explicitly in the discussion.

Minor

1. It would be nice to assess if the Ipo13b mutants exhibited deficits in escape response, as would be expected if Mauthner cell conduction velocity was slowed.
2. Panels in Figure 3L and M were cutoff.

Reviewer #3 (Remarks to the Author):

This manuscript by Jenea Bin et al. succeeds in establishing zebrafish as a new model system to study the mechanisms of axonal diameter regulation, a scientifically very interesting and so far not well understood topic with relevance for behavioral capabilities. In a screen for mutations affecting the diameters of the exceptionally large-diameter Mauthner axons, the authors find that the function of the nuclear receptor Importin 13b is required for Mauthner axons to developmentally gain their normal diameters, a highly interesting and novel observation. Importantly, the relevance of Importin 13b is supported by targeted mutagenesis in additional zebrafish lines. The specificity of the effect for axonal diameter is underlined by normality of axon length and cell body size. It is a striking finding that nerve conduction velocity correlates with axonal diameters. Moreover, reduced axonal diameters correlate with – and are probably explained by - reduced neurofilament density and number. There are many interesting aspects in this manuscript; yet among them the neuron specific mutant (Figures 6-7) may be most relevant because it proves that the observed effect is at least in part neuron-intrinsic. Together, the work is interesting, relevant and convincing, the text is well written, the conclusions are justified by the data, and I have just a few observations that the authors may want to consider before publication.

Major point

Unless I have overlooked it - and if so I apologize - there is no assessment of the myelination status of Mauthner axons in Ipo13b-ue57 and the neuron-specific mutants. I agree that it is quite reasonable to assume that they are myelinated. However, considering the key relevance for the manuscript, the argument that the reduced NCV is caused by reduced axonal diameters may be further strengthened by assessing Mauthner axon myelination status using either electron or light microscopy

Minor points

Title: the authors may want to consider specifying Importin 13 as Importin 13b

Abstract line 7: the authors may want to consider specifying "overall cell size" as cell body size or soma size

Fig 1D: The authors may want to consider showing individual datapoints, additional to mean and error. In case the number of datapoints is too large to display, violin plots may be an option. It is not clear from the legend how many samples were assessed and if SEM or SD is shown

Fig 2C and 2J: Do I understand correctly that these plots essentially show the same assessment, but one time with GFP and one time with RFP as marker?

Mauthner axon diameters are reduced by about half in Ipo13b-ue57 mutants and roughly by 25% in the neuron-specific mutants. The authors may want to consider extending their discussion by a speculation if projection targets (see ref 15) or Mauthner-associated glial cells account for the other 25%, i.e. the possibility that other, non-neuronal cells depend on Importin 13b expression for their function(s) in regulating Mauthner axon diameters. For example, it may be interesting to assess Mauthner axon diameters in myelinating glia-specific Importin 13b mutants. However, this idea is beyond the scope of the present study and thus could rather be the topic of follow-up work and is not mandatory for the present manuscript

Reviewer #4 (Remarks to the Author):

The regulation of axon diameter is physiologically significant but remains poorly understood. In this study, Bin et al report that a mutation of importin 13b results in smaller axon diameters in zebrafish, notably for the large myelinated axon of Mauthner neurons, whereas their soma size is normal. They then show this mutation appears to be primarily neuron-specific based on a Crispr mutation that impacts neurons. However, as they note they cannot rule out mosaic and/or non-autonomous effects since the reduction of axon diameter in these Crispr generated mutants is partial. Finally, based on physiological analyses, they provide evidence that reduced axon diameter results in a predicted reduction of conduction velocity whereas it had no impact on high frequency firing or jitter/axon conduction fidelity.

The paper is well written and while the data shown is convincing, limitations in their analysis reduce the impact of the findings. One issue is that the study does not substantially advance our understanding of how axon diameter is regulated as importin 13 has pleiotropic effects on protein/transcript expression (underscored by the lethality of the mutant). The authors note that elucidating the specific mechanisms involved is likely to be a difficult task given the large numbers of proteins/pathways that may be impacted. Based on EM analysis, NF packing density is reduced and NF abundance is likewise calculated to be reduced. This is a common endpoint in axons with reduced diameters but perhaps additional studies could make this link to axon diameter stronger. As it stands, we are left with an interesting but enigmatic result with limited insight into how this mutation affects axon diameter.

The authors next show the impaired diameter affects conduction velocity to the level of that seen in smaller wt axons at earlier ages. This is an expected result as the diameter of myelinated axons is well known to be a primary determinant of conduction velocity. While it could be argued the experimental approach here is distinct from earlier studies that compared axons of the same age with differing diameters, the conceptual advance of their findings is nominal. Here, they focus exclusively on axon diameter as the key determinant. Plausibly other parameters that may co-vary with diameter (internode length, myelin sheath thickness, node length, etc) could contribute to reduced velocities but have not been measured. Hence, it is difficult in the absence of other morphological parameters to conclude the changes are principally/exclusively the result of reduced axon diameter

Minor comments:

While the paper does a good job in referencing some of the older literature on the regulation of axon diameter, there is a significant literature that implicates neurofilament numbers and phosphorylation in the regulation of axon diameter - and in some reports - ion channels (see Yuan et al CSH Perspect Biol 2017 Apr; 9(4): a018309 for review). This literature would seem mechanistically relevant and they may wish to cite

There is a very minor editing mistake ("of "is missing) in the first paragraph of the discussion i.e., "...axon outgrowth or growth of the neuron as a whole."

We thank the reviewers for their positive assessment of our manuscript and their very thoughtful comments and suggestions. Upon reviewers' comments, we have significantly revised our manuscript, including adding new datasets:

1. We carry out further characterization of neuron-specific importin 13b mutants.

We performed detailed morphological analyses of Mauthner neurons in neuron-specific mutants and find that while their growth in axon diameter is impaired, their axons grow to a normal length, and their cell bodies and dendrites exhibit normal growth. This underscores the specificity of the role of importin 13b in regulating axon diameter growth.

2. We characterize myelination of the Mauthner axon over time and in neuron-specific importin 13b mutants.

The importance of assessing myelin along axons with reduced diameter was duly noted, and we carried out an extensive series of analyses of myelin around the Mauthner axon by super-resolution live imaging of controls and neuron-specific *ipo13b* mutants. This revealed that the smaller diameter Mauthner axons in neuron-specific *ipo13b* mutants are still myelinated along their length and that their myelin grows to the same thickness seen in controls. This latter observation indicates that the growth of myelin in thickness does not occur in direct response to changes in axon diameter over this time. This observation helped refine the interpretation of our assessment of conduction properties in neuron-specific *ipo13b* mutants, and supported our general conclusion that axon diameter growth plays a major role in driving increases to conduction velocity along the Mauthner axon that occur over time.

3. We have significantly revised the manuscript throughout to include our new data and to clarify our assessment of how axon diameter growth influences action potential conduction. In particular, our new data showing that myelin thickness is the same on smaller diameter axons in neuron-specific *ipo13b* mutants allowed us to substantiate our initial conclusion that axon diameter growth is a major driver of the increases in conduction velocity seen along the Mauthner axon over time.

We believe that the manuscript has been greatly improved by taking reviewer feedback on board, and thank the reviewers in advance for further consideration.

Point by point responses to reviewers' comments are interspersed below.

REVIEWER COMMENTS

Reviewer #1 (Remarks to the Author):

Bin et al identify importin-13b (*ipo13b*) as a regulator of axon diameter and action potential conduction speed. A diverse set of experimental strategies were used to describe the impact of *ipo13b* loss and functional consequences on neuronal physiology. EM was used to show that *ipo13b* broadly affects axon diameters throughout the spinal cord. The authors also take a more focused approach using the Mauthner neuron, allowing reliable characterization of a single identifiable neuron, and perform super resolution imaging and in vivo electrophysiology to show detailed structural and functional consequences in *ipo13b* mutants or following neuron-specific knockdown through development, which provided complementary results. Structural analysis demonstrates that *ipo13b* is critical for the growth of axon diameter during development, yet not other structural features of neurons. Functional experiments, using electrophysiology, also demonstrate specific effects, whereas only conduction speed is impacted yet not the ability to entrain sustained stimulation. Collectively, this is a very nice study that implements many of the strengths of the zebrafish system, forward genetics, optical imaging, electrophysiology etc., to uncover novel mechanisms regulating fundamental properties of nervous system function. Genetic regulation of axon diameter is poorly understood and uniquely challenging to address, which makes this current study highly impactful and likely to be of broad interest. The experiments and analysis are rigorous and provide strong support to the conclusion drawn. The manuscript is appropriate for publication in Nature Communications, yet there are several points which I think would help strengthen conclusions and clarify interpretations.

We thank the reviewer for their positive assessment of our manuscript, and hope that we have sufficiently addressed their points below.

1. The authors use multiple CRISPR alleles to recapitulate the screen mutant phenotype as evidence that *ipo13b* is the basis of the observed axon diameter phenotypes. A rescue experiment would provide strong supporting evidence. An ideal experiment would be mosaic rescue in 1 but not both Mauthner neurons. However, with the strength of the evidence already provided any form of rescue experiment would be a solid addition.

We thank the reviewer for this suggestion. We have now added an experiment in which we show that the axon diameter phenotype in the *ipo13b*^{ue57} mutants can be rescued by the injection of *ipo13b* mRNA at the single cell stage (Figure 2J), further supporting the conclusion that *ipo13b* is the gene responsible for the axon diameter phenotypes.

As the reviewer suggested, it would be ideal to mosaic rescue *ipo13b* in one, but not both, Mauthner axons. We have previously attempted these experiments, however transgenic overexpression of *ipo13b* in neurons led to variable outcomes, including toxicity, which confounded potential analyses. Based on these early observations, we speculated that the amount of *ipo13b* expressed in neurons is tightly regulated, and that our transgene resulted in a level of *ipo13b* expression that is not tolerated by neurons, and so we did not pursue this approach further.

As the reviewer notes though, we have generated additional mutant alleles disrupting *ipo13b*, and studied those in homozygous and trans-heterozygous forms, all of which resulted in reduced axon diameter. We are entirely confident in our finding that disruption to *ipo13b* leads to impaired growth of axons in diameter.

2. *Ipo13b* mutants show no change in Mauthner cell body area, yet volume or at least a measure from an orthogonal view would strengthen the claim that *ipo13b* does not impact cell body growth. Similarly, quantifying the length, or any other parameter, of the lateral dendrite would help support that *ipo13b* is highly specific to regulating axon diameter.

We thank the reviewer for this suggestion and have added an extensive new dataset to address this important point. Specifically, we have extended our analyses to now include measurements of the cell body and lateral dendrite over time in the neuron-specific *ipo13b* mutants and show that they both continue to grow normally over a period during which axon diameter growth is disrupted (Figure 6 F & G). We have also measured the volume of the Mauthner cell body in z-stack images and shown that this is not affected in neuron-specific *ipo13b* mutants (Figure 6H). We believe this data strengthens our claim that *ipo13b* does not impact cell body or dendrite growth and is highly specific to regulating axon diameter.

3. Some additional measures from the physiological recordings would be helpful. For example, action potential rise time and half-width.

The reason we have not included measures related to the action potential waveform (e.g. rise time, half-width) is due to the configuration of our recordings. We are currently restricted to assessing conduction properties of action potentials travelling in the antidromic configuration, such that we are measuring the arrival of action potentials to the cell body in response to axonal stimulation. Since the Mauthner neuron has a very low input resistance, the current-clamp does not extend sufficiently to allow us to measure the action potential waveform in the axon. Therefore, the recorded waveform properties are somatic and not informative as to the electrical properties of the axon, which is the focus of our paper. We would prefer to maintain the focus of our functional analyses on how changes to axon diameter affect conduction along the axon. Importantly, we note to the reviewer that there are no changes to the somatic action potential waveform that would be consistent with any skewing of our conduction velocity data. We hope this clarifies why we have not included action potential wave form properties (rise time and half-width) within the manuscript.

4. Some discussion is provided regarding how importin 13b may be modulating axon diameter and the authors do a good job at highlighting how challenging this mechanism may be to resolve. However, this discussion is lean and it would be rather interesting to expand upon putative ways importin 13b may be regulating axon diameter.

We have generally expanded our discussion, including to highlight different ways in which importin 13b may be modulating axon diameter, for example via cell-intrinsic growth pathway or in response to cell-extrinsic signals, as well as the possibility of non cell-

autonomous roles in other neurons. We would be happy to consider further points of discussion that the reviewer may want to raise.

Minor points:

1. In Figure 5, the changes to neurofilament density is interesting, yet the data and interpretation is a bit confusing (eg increased measures in mutants – Figure 5C, yet overall this is a decrease due to the decreased axon volume). It may be worth seeing the images Figure 5A-B with matched as well as the currently shown adjusted scales. Potentially an additional figure where the data is scaled to volume to provide a clear visual representation of the overall decrease in neurofilaments.

We thank the reviewer for this comment and apologize for the confusion. There was an error in the dimensions of the scale bars noted in the figure legend, which has now been corrected. Both the control and *ipo13b^{ue57}* mutant images in Figure 5 A-B are at matched scales. We have also modified some of the wording used in the results section to describe the changes to neurofilaments, which we hope makes these results and their interpretation clearer to the reader.

2. First results section the authors state “project their axons down the entire length of the contralateral spinal cord (Figure 1B)”. Figure 1B as it is does not show the projection. I suggest either adding an additional image or simply add a citation for clarity.

Thank you for this suggestion, we have now added an image of the entire Mauthner neuron to Figure 1 (panel D), as well as a citation to the main text.

3. Figure 2D the authors call the behavioral result “display reduced motility”, yet this is somewhat confusing. The 5 dpf larvae spend less time moving, which may be more accurate as reduced time swimming or fewer swim initiations. Conversely, when swimming is engaged the overall motility (distance, velocity, etc) could be normal or reduced. No experiments are needed yet a text change would clarify.

Thank you for this suggestion. We appreciate that “reduced motility” can be interpreted in different ways. We have replaced this figure panel with two additional graphs showing that *ipo13b^{ue57}* mutants initiate fewer swim bouts, but that these swim bouts are longer in duration. Overall, this results in less time spent moving during the 20 min open field test.

4. Early in development mutant axon diameters are comparable to siblings. Are early behaviors similarly comparable?

This is a great question, but unfortunately, at the time points when axon diameter is normal (<3 dpf), zebrafish larvae have not yet hatched from their chorions; therefore, we cannot assess swimming behaviour.

5. Do the authors think motoneurons are impacted by the loss of *ipo13b*? Experiments are not needed, yet may be an interesting point of discussion and potential role of *ipo13b* in the peripheral nervous system.

This is another important question. We do see that axon diameter is affected in the peripheral nervous system in the full *ipo13b* mutants. However, these mutants also have a severe reduction in the number of Schwann cells, meaning many axons that are normally myelinated are either unmyelinated or partially myelinated. Since we know Schwann cells are regulators of axon diameter in the PNS, it is difficult to disentangle whether the reduction in diameter is due to disruption of *ipo13b* or disruption to myelin in full mutants. We have not looked at the PNS in our neuron-specific *ipo13b* mutants; however, we expect this would be challenging due to the mosaicism in the CRISPR/Cas9 based model used in this study. This could be something we address in the future, by using recently published improved techniques in zebrafish that can disrupt gene function in a cell-type and temporal specific manner without the mosaicism inherent to the CRISPR/Cas9 based cell-type specific strategy used in this study. Interestingly, and for the reviewer's insight, we are currently in the process of putting together a manuscript which shows that oligodendrocytes/myelin in the CNS does not affect axon diameter growth (both in zebrafish and mice). This divergence in the mechanisms regulating axon diameter in the CNS and PNS is one of the reasons we have chosen to only focus on the CNS in our paper.

6. In the results expanding on what jitter is and why it is an important metric would be helpful to a reader

We thank the reviewer for this suggestion. We have expanded on this within the results section.

Reviewer #2 (Remarks to the Author):

This manuscript describes the results of a mutagenesis screen in zebrafish to identify genes that influence axon diameter. Using an ENU forward genetics screen, they map one mutant displaying reduced diameter of Mauthner cell axons to the importin 13b locus. Importin 13 is a chaperone protein that assists in moving a variety of proteins, including transcription factors, from the cytoplasm to the nucleus and has been shown to play a role in neuronal development and synaptic function. Validation is solid, as two CRISPR mutants at different sites in the gene exhibit similar growth deficits, and the reduction in axon diameter was also observed in other neurons in the whole body *lpo13bue57* mutant, with the largest decline seen in larger diameter axons. The decline in axon diameter growth was accompanied by a reduction in overall myelin thickness, although this was not quantified. *lpo13* is however, widely expressed throughout the body, so complex interactions could be responsible for this phenotype. To address this, the authors show using neuronal specific CRISPR deletion that neuron-specific loss accounts for the most of the change in axon diameter.

Using cell attached patch recordings of back propagating action potentials from Mauthner cells, the authors find *lpo13bue57* mutants exhibit markedly slower conduction velocity. Control and mutant fish at a stage when the axons have a similar diameter had similar conduction velocities, suggesting that it is likely the change in axon diameter, rather than other possible aspects of *lpo13b* function that are responsible for determining the speed of AP propagation.

Together, these studies introduce fluorescent labeling of Mauthner cells as a means to explore the molecular mechanisms that control axon diameter and identify importin13b as a key upstream regulatory protein in this process. Further, the careful physiological measurements provide evidence that axon diameter is a major determining factor in controlling the conduction velocity of action potentials.

The text is written with exceptional clarity and brevity, the results were analyzed rigorously, and the figures well illustrated. I have only a few suggestions for improvement.

We thank the reviewer for their positive assessment of our manuscript, and hope that we have sufficiently addressed their points below.

Major

1. Given the importance of myelination on conduction velocity, it is surprising that the studies do not provide quantitative measures of myelination that coincide with the axon diameter changes. Ideally, one would like to see Mauthner cell g-ratio plots across dpf, although it does not appear from the electron microscopy images provided that myelin was sufficiently persevered. I am not an expert in zebrafish EM, so maybe this type of analysis isn't possible. If not, a light level estimation of myelin thickness across age and genotype would be helpful.

Thank you for this suggestion. We have added extensive characterization of myelination along the Mauthner axon to the manuscript, including quantification to show that the Mauthner axon is myelinated along its length at the time points at which we perform our electrophysiology experiments in both control and neuron-specific *ipo13b* mutants (Figure 1 F & G, Figure 7F). Furthermore, we have quantified myelin thickness/g-ratio over time in controls, and in 5 dpf neuron-specific *ipo13b* mutants. We agree with the reviewer that this is an important addition to the paper, as previous studies have shown that there is a correlation between axon diameter and myelin thickness when comparing different axons, but few studies have actually looked at how myelin thickness changes along axons as they grow in diameter over time. As the reviewer pointed out, myelin preservation in zebrafish electron microscopy is challenging, and also does not allow us to follow individual axons over time, so we assessed myelin thickness/g-ratios using super-resolution confocal live-imaging, as per the reviewers suggestion (refer to Supplemental figure 1 and material and methods).

We find, as expected, that myelin along control Mauthner axons grows in thickness between 3 dpf and 5 dpf, with a small, but significant, decrease to the g-ratio (Figure 7 A – E). Interestingly however, myelin thickness is comparable between control and neuron-specific *ipo13b* mutant Mauthner axons at 5 dpf, despite neuron-specific *ipo13b* mutants having significantly reduced axon diameters. This result was also verified as best as possible by electron microscopy (Figure 7G-N, Supplementary figure 2). This indicates that the growth of myelin in thickness over this time period is not in direct response to changes in axon diameter. We discuss this finding in and of itself, and also note that this underscores even further the important role that axon diameter growth plays in determining the increases in conduction velocity that are seen along the Mauthner axon over time, since conduction velocities in neuron-specific *ipo13b* mutants are more tightly linked to axon diameter, than increases to myelin thickness or the developmental age of the neuron.

2. The description and analysis of the neuron specific Ipo13b mutants is a bit stunted. The authors suggest that differences between complete and neuron specific deletions “reflect the timing or extent of loss of importin 13b function, or additional non-cell autonomous contributions of importin 13b.” It would be helpful to include quantifications of axon diameter in the text, so that the reader doesn’t have to estimate from the figure (in general, this would be helpful for many of the experiments, as there few statistics or statistical tests included in the text). A limitation here is that there are no data to assess various possibilities, so one is left wondering what accounts for the difference between complete and neuron specific deletions. Immunocytochemistry against Importin 13b would be beneficial to the study and would help address these possibilities.

We thank the reviewer for highlighting the need to clarify the comparison of full and neuron-specific *ipo13b* mutants further. As suggested, we have added to the results text a comparison of the % decrease to axon diameter observed in neuron-specific *ipo13b* mutants compared to the % decrease to axon diameter observed in full *ipo13b* mutants, and expanded on the reasons why neuron-specific *ipo13b* mutants might be less severe. We agree that immunocytochemistry against importin 13b would be helpful to assess which cell types express importin 13 to be able better predict non-cell autonomous roles. Unfortunately, antibodies targeting importin 13 do not work well in zebrafish. It is important to note though that for our functional analyses, the differences between full and neuron-specific *ipo13b* mutants do not affect the interpretation of our results, as we are simply using the neuron-specific *ipo13b* mutants as a tool to reduce axon diameter.

We also thank the reviewer for highlighting the need to provide more detailed description of the neuron-specific *ipo13b* mutants. In addition to the myelin data noted above, we have also added a new and extended analyses of neuronal and axon morphology in neuron-specific *ipo13b* mutants, which show that cell body and dendrite size, as well as axon length, are unaffected. This further points to the specificity of *ipo13b* function in neurons in supporting axon diameter growth.

3. One of the main conclusions of the study is that the increases in conduction velocity along the myelinated Mauthner axon are primarily driven by its growth in diameter. This is based on the observation that control and knockout axons of the same diameter exhibited similar conduction velocities. I feel that this conclusion oversimplifies considerably, as it is possible that importin13b controls the expression of many genes that together coordinate axonal maturation. If genes encoding ion channels, transporters, synaptic release (target derived growth factor sensitivity), etc. are co-regulated by importin13b (for example, by a key transcription factor), then their expression could also scale with diameter. The implication from the text is that axon diameter is driving these changes in functional properties, but there is limited evidence in the study to test this hypothesis. It seems more likely that many aspects of axonal maturation are co-regulated by Ipo13b. This possibility should be addressed more explicitly in the discussion.

We agree with the reviewer’s point and should have made this more explicit in our original submission. We now add this point to our discussion, noting that it is entirely possible that importin 13b could co-regulate several features of axonal maturation in addition to axon

diameter, such as expression of ion channels, as the reviewer notes. We have also, we hope, clarified our conclusions based on the analysis of conduction velocity along control axons over time (3,4 and 5 dpf) and how this relates to conduction velocity along neuron-specific *ipo13b* mutant axons at 5 dpf.

We find that *ipo13b* mutant axons at 5 dpf conduct action potentials at a velocity that is tuned to their axon diameter, matching conduction velocities seen along control axons of the same size at earlier stages. This points to a general disconnect between the developmental maturation of the neuron and the increases to conduction velocity observed over time, and indicates that the latter is tightly linked to axon diameter growth. If other parameters that mature independently of diameter were significantly contributing to the increases in conduction velocity, we would have observed divergence from the relationship between axon diameter and conduction speeds measured in controls. Our new analysis showing that maturation to myelin thickness still occurs normally over this time period, such that both controls and neuron-specific *ipo13b* mutants have the same thickness of myelin despite significantly different axon diameters, further supports a model that it is axon diameter growth that best predicts increases in conduction velocity.

Although we cannot rule out a specific co-regulation of axon diameter growth and ion channels (for example) by *ipo13b* (as we now elaborate on in the discussion), our investigation found that disruption to *ipo13b* did not compromise other crucial aspects of conduction, including the ability to sustain high frequency firing and the precision of action potential conduction. These conduction features are intricately linked to the expression and localisation of ion channels along the axon (and other parameters), challenging the notion of a simple co-regulation of axon diameter and general maturation of the functional axolemma. We hope that we have made these points sufficiently clear in the revised manuscript, but would welcome any further refinements that the reviewer might have.

Minor

1. It would be nice to assess if the *Ipo13b* mutants exhibited deficits in escape response, as would be expected if Mauthner cell conduction velocity was slowed.

This is a good suggestion. However, the majority of full *ipo13b* mutants do not inflate their swim bladders, making refined high-speed kinematic analyses of the escape response very difficult. The neuron-specific *ipo13b* mutants are mosaic in the nature of the mutations that are induced within individual neurons, as noted and discussed above, which precludes robust measurements of whole animal behaviours such as the escape response that rely on several neurons, not just the Mauthner cell.

2. Panels in Figure 3L and M were cutoff.

We have double checked these figure panels and trust that they now appear properly.

Reviewer #3 (Remarks to the Author):

This manuscript by Jenea Bin et al. succeeds in establishing zebrafish as a new model system to study the mechanisms of axonal diameter regulation, a scientifically very interesting and so far not well understood topic with relevance for behavioral capabilities. In a screen for mutations affecting the diameters of the exceptionally large-diameter Mauthner axons, the authors find that the function of the nuclear receptor Importin 13b is required for Mauthner axons to developmentally gain their normal diameters, a highly interesting and novel observation. Importantly, the relevance of Importin 13b is supported by targeted mutagenesis in additional zebrafish lines. The specificity of the effect for axonal diameter is underlined by normality of axon length and cell body size. It is a striking finding that nerve conduction velocity correlates with axonal diameters. Moreover, reduced axonal diameters correlate with – and are probably explained by - reduced neurofilament density and number. There are many interesting aspects in this manuscript; yet among them the neuron specific mutant (Figures 6-7) may be most relevant because it proves that the observed effect is at least in part neuron-intrinsic. Together, the work is interesting, relevant and convincing, the text is well written, the conclusions are justified by the data, and I have just a few observations that the authors may want to consider before publication.

We thank the reviewer for their positive assessment of our manuscript, and hope that we have sufficiently addressed their points below.

Major point

Unless I have overlooked it - and if so I apologize - there is no assessment of the myelination status of Mauthner axons in *Ipo13b-ue57* and the neuron-specific mutants. I agree that it is quite reasonable to assume that they are myelinated. However, considering the key relevance for the manuscript, the argument that the reduced NCV is caused by reduced axonal diameters may be further strengthened by assessing Mauthner axon myelination status using either electron or light microscopy.

Thank you for this important suggestion. We have added extensive characterization of myelination along the Mauthner axon to the manuscript, including quantification to show that the Mauthner axon is myelinated along its length at the time points at which we perform our electrophysiology experiments in both control and neuron-specific *ipo13b* mutants (Figure 1 F & G, Figure 7F). We have focussed on analyses of myelination in the neuron-specific *ipo13b* mutants because these are the mutants we use to study the relationship between axon diameter and conduction velocities.

As detailed in the revised manuscript, we have quantified myelin thickness/g-ratio of myelin along the Mauthner axon over time in controls, and in 5 dpf neuron-specific *ipo13b* mutants to match the timing of our functional studies. Previous studies have shown that there is a general correlation between axon diameter and myelin thickness when comparing different

axons, but few studies have actually looked at how myelin thickness changes along axons as they grow in diameter over time.

We find that myelin along control Mauthner axons grows in thickness between 3 dpf and 5 dpf, with a small, but significant decrease to the g-ratio (Figure 7 A – E). Interestingly however, myelin thickness is comparable between control and neuron-specific *ipo13b* mutant Mauthner axons at 5 dpf, despite neuron-specific *ipo13b* mutants having significantly reduced axon diameters. This results in mutant Mauthner axons having smaller g-ratios than controls (Figure 7G-N, Supplementary figure 2). This indicates that the growth of myelin in thickness over this time period is not in direct response to changes in axon diameter. We discuss this finding in and of itself, and also note that this underscores even further the important role that axon diameter growth plays in determining the increases in conduction velocity that are seen along the Mauthner axon over time, since conduction velocities in neuron-specific *ipo13b* mutants remain tightly linked to axon diameter, despite increases to myelin thickness and developmental age of the neuron.

Minor points

Title: the authors may want to consider specifying Importin 13 as Importin 13b

We would rather keep the protein name more general for the title, but can adapt if there is a consensus view that we should.

Abstract line 7: the authors may want to consider specifying “overall cell size” as cell body size or soma size

This has been amended as suggested.

Fig 1D: The authors may want to consider showing individual datapoints, additional to mean and error. In case the number of datapoints is too large to display, violin plots may be an option. It is not clear from the legend how many samples were assessed and if SEM or SD is shown

We apologize that the sample size was omitted from the figure legend. We have now amended this figure panel to show individual data points. As stated in the material and methods, all data is presented as means with standard deviation unless otherwise indicated.

Fig 2C and 2J: Do I understand correctly that these plots essentially show the same assessment, but one time with GFP and one time with RFP as marker?

We thank the reviewer for pointing this out. We have now removed Figure 2J.

Mauthner axon diameters are reduced by about half in *lpo13b-ue57* mutants and roughly by 25% in the neuron-specific mutants. The authors may want to consider extending their discussion by a speculation if projection targets (see ref 15) or Mauthner-associated glial cells account for the other 25%, i.e. the possibility that other, non-neuronal cells depend on

Importin 13b expression for their function(s) in regulating Mauthner axon diameters. For example, it may be interesting to assess Mauthner axon diameters in myelinating glia-specific Importin 13b mutants. However, this idea is beyond the scope of the present study and thus could rather be the topic of follow-up work and is not mandatory for the present manuscript

We thank the reviewer for these thoughtful comments. We have extended the text within the results and discussion to include speculation about possible non-cell autonomous roles for importin 13b, and the importance in extending cell-type specific studies in the future.

Reviewer #4 (Remarks to the Author):

The regulation of axon diameter is physiologically significant but remains poorly understood. In this study, Bin et al report that a mutation of importin 13b results in smaller axon diameters in zebrafish, notably for the large myelinated axon of Mauthner neurons, whereas their soma size is normal. They then show this mutation appears to be primarily neuron-specific based on a Crispr mutation that impacts neurons. However, as they note they cannot rule out mosaic and/or non-autonomous effects since the reduction of axon diameter in these Crispr generated mutants is partial. Finally, based on physiological analyses, they provide evidence that reduced axon diameter results in a predicted reduction of conduction velocity whereas it had no impact on high frequency firing or jitter/axon conduction fidelity.

The paper is well written and while the data shown is convincing, limitations in their analysis reduce the impact of the findings. One issue is that the study does not substantially advance our understanding of how axon diameter is regulated as importin 13 has pleiotropic effects on protein/transcript expression (underscored by the lethality of the mutant). The authors note that elucidating the specific mechanisms involved is likely to be a difficult task given the large numbers of proteins/pathways that may be impacted. Based on EM analysis, NF packing density is reduced and NF abundance is likewise calculated to be reduced. This is a common endpoint in axons with reduced diameters. As it stands, we are left with an interesting but enigmatic result with limited insight into how this mutation affects axon diameter.

The authors next show the impaired diameter affects conduction velocity to the level of that seen in smaller wt axons at earlier ages. This is an expected result as the diameter of myelinated axons is well known to be a primary determinant of conduction velocity. While it could be argued the experimental approach here is distinct from earlier studies that compared axons of the same age with differing diameters, the conceptual advance of their findings is nominal. Here, they focus exclusively on axon diameter as the key determinant. Plausibly other parameters that may co-vary with diameter (internode length, myelin sheath thickness, node length, etc) could contribute to reduced velocities but have not been measured. Hence, it is difficult in the absence of other morphological parameters to conclude the changes are principally/exclusively the result of reduced axon diameter

We thank the reviewer for their series of important points, which highlighted that we needed to clarify the novelty and importance of our core observations, and also to acknowledge where further work beyond the remit of the manuscript will be required.

In our revised manuscript we have included a much more detailed analysis of neuronal and axonal morphology in neuron-specific *ipo13b* mutants, which unlike full mutants live a normal lifespan, as now noted in the revised manuscript. We find that while *ipo13b* is required for axons to grow in diameter, it is not required for cell growth in general, with no changes observed in the size of the cell body or lateral dendrite, or in axon length. This provides further evidence that *ipo13b* plays an important and specific role in regulating axon diameter growth, which provides a molecular entry point to understand the mechanisms controlling axon diameter growth in the CNS. As the reviewer notes, our understanding of axon diameter growth as a field has been fairly focussed on the cytoskeleton, and in particular the roles of neurofilaments. We see the establishment of zebrafish as a model to study axon diameter as a major feature of this manuscript. We fully acknowledge that we are only scratching the surface from a mechanistic perspective, but hope the reviewer agrees that this model, including the identification of *ipo13* as an entry-point, will provide an important way in which we can advance our understanding of the biology of axon diameter growth.

The reviewer is, of course, entirely correct in pointing out that a relationship between axon diameter and conduction velocity has been clear for quite some time, and we see in hindsight that we did not highlight the novelty of our findings sufficiently well in our original submission. In our study, we have focussed on the growth of single axons in diameter over time, rather than comparing axons of different diameters. During development, myelinated axons have the potential to use multiple strategies to increase their conduction speed, including changes to diameter, myelin, nodes of Ranvier, ion channels etc. In our revision we hope that we have clarified our conclusions that growth of axons in diameter is a major driver of the increases in conduction velocity that emerge over time along myelinated axons. Our conclusion is based on the analysis of conduction velocity along control Mauthner axons over time (3,4 and 5 dpf) and how this relates to conduction velocity along neuron-specific *ipo13b* mutant axons at 5 dpf. We find that *ipo13b* mutant axons at 5 dpf conduct action potentials at a velocity that is tuned to their axon diameter, and that this matches conduction velocities seen along control axons of the same size at earlier stages. This points to a general disconnect between the developmental maturation of the neuron and the increases to conduction velocity observed over time, and indicates that the latter is tightly linked to axon diameter growth. If other parameters that mature independently of diameter were significantly contributing to the increases in conduction velocity, we would have observed divergence from the relationship between axon diameter and conduction speeds measured in controls. Our new analysis showing that maturation to myelin thickness still occurs normally over this time period, such that both controls and neuron-specific *ipo13b* mutants have the same thickness of myelin despite significantly different axon diameters, further supports a model that it is axon diameter growth that best predicts increases in conduction velocity. Although we cannot rule out a specific co-regulation of axon diameter growth and ion channels (for example) by *ipo13b* (as we now elaborate on in the discussion), our investigation found that disruption to *ipo13b* did not compromise other crucial aspects of conduction, including the ability to sustain high frequency firing and the precision of action potential conduction. These conduction features are intricately linked to the expression and localisation of ion channels along the axon (and other parameters), challenging the notion of a simple co-regulation of axon diameter and general maturation of the functional axolemma. We hope that we have made these points sufficiently well in

the revised manuscript, but would welcome any further refinements that the reviewer might have, so that this is clear for readers.

We absolutely agree with the reviewer that as a field we need to move towards a much more holistic view of the myelinated axon, and that we need to deconstruct the relative influence of the various factors that influence distinct aspects of conduction. We have added significantly to our discussion to address this point. It is clear that working towards understanding the mechanisms that regulate the many features of the myelinated axon that influence conduction such as growth in diameter, the expression, delivery and localisation of ion channels and transporters to correct axonal domains, myelination, and more will require a huge amount of targeted investigation by the community. We hope that our study makes the point that an often overlooked aspect of myelinated axons, which is their growth in diameter, is an important component to consider in moving towards such a holistic understanding of how conduction properties mature during development, are refined throughout life, and are affected in disease.

Minor comments:

While the paper does a good job in referencing some of the older literature on the regulation of axon diameter, there is a significant literature that implicates neurofilament numbers and phosphorylation in the regulation of axon diameter - and in some reports - ion channels (see Yuan et al CSH Perspect Biol 2017 Apr; 9(4): a018309 for review). This literature would seem mechanistically relevant and they may wish to cite.

We thank the reviewer for pointing us to this review and to the relevant literature cited within, highlighting the diameter-independent roles of neurofilaments in modulating ion channels and conduction. We have now included these important points in our modified discussion.

There is a very minor editing mistake ("of "is missing) in the first paragraph of the discussion i.e., "...axon outgrowth or growth of the neuron as a whole."

Thank you. This has been corrected in the manuscript.

REVIEWERS' COMMENTS

Reviewer #1 (Remarks to the Author):

The authors have addressed my concerns and in my opinion the manuscript is ready for publication. This is a rigorous and exciting study.

Reviewer #1 (Remarks on code availability):

Acceptable

Reviewer #2 (Remarks to the Author):

The authors have conscientiously addressed the concerns raised in the prior review, by performing additional quantification of the structural features of Importin13 deficient neurons and of myelin thickness. They have also expanded the discussion to speculate about the possible mechanisms responsible for the change in axon diameter in the mutants. These changes have significantly improved the study. My only lingering concern relates to their firm conclusion that axon diameter is the primary driver of the changes in conduction velocity observed. Since they do not provide mechanistic insight into the pathways regulated by Importin13 and this protein has known pleiotropic effects, this seems a likely oversimplification. That is, from the data provided it does not seem convincing that the observed effects can be attributed solely to changes in axon diameter as a physical property - related only to changes in capacitance and internal resistance. It seems reasonable that there may be many co-regulated genes that control conduction velocity (eg. ion channel density and composition, internode length, etc), and possible complex interactions between ion channel expression and axon diameter growth. Without manipulation of other possibilities, I think that this conclusion should be softened. From the response, I think that this could mostly be a semantic issue, but nevertheless one that could be confusing to readers. I suggest that the authors take a more neutral stance in the title and abstract about the mechanisms responsible for the conduction velocity change, perhaps by using the phrase "axon diameter and co-regulated processes" or something similar. Of note, I think the evidence that myelination does not appear to adapt to this reduction in diameter is fascinating and should be further highlighted in the abstract, as this runs counter to the prevailing evidence of axon diameter being a primary driver of myelin thickness.

Reviewer #2 (Remarks on code availability):

This link was not associated with an active page on Github, so I was unable to review the code.

Reviewer #3 (Remarks to the Author):

The authors have adequately considered my comments. In my view the manuscript is fit for publication.

Reviewer #4 (Remarks to the Author):

This is a resubmission of a study that compellingly demonstrates importin 13 is required for radial expansion of large CNS axons in zebrafish while sparing the soma and dendrites. They further show the increase in axonal diameter is the major driver of increased conduction velocity independent of myelin thickness. The authors have been responsive to concerns raised in previous reviews including providing a rescue experiment for the mutant, characterizing myelination more completely, and revising the text/discussion. These changes have strengthened the manuscript. While the mechanisms by which importin 13 regulates axon diameter are unclear, the authors reasonably argue elucidation is beyond the scope of the current study.

I have one modest, lingering concern and otherwise minor issues the authors should address:

The evidence that conduction is dependent on axon diameter and not myelin thickness is mostly convincing. However, other parameters that were not examined - including node width, internode length, and paranode integrity have all been implicated in regulating conduction velocity as the authors discuss. In particular, as axon diameter has been shown to be a determinant of internode length (Bechel et al, 2015) - which in turn impacts CV - this parameter seems particularly relevant to measure in their mutant, perhaps even from existing images. This data would also shed physiological insights into whether myelin thickness is indeed regulated by axon diameter in contrast to myelin thickness.

The authors show a graph quantifying rescue of axon diameter in Fig. 2J; it would be useful to show a micrograph of an example of the rescued axon as well in the figure

Do they know if the original C-terminal ue57 mutant affects importin 13 function or does it primarily affect its abundance?

The authors demonstrate there is no change in myelin thickness in the neuron-specific knockout. I may have missed it but is myelin thickness also unchanged in the original pan-mutant?

The diameter of the Mauthner axon tapers from proximal to distal based on examination of Supplemental Figure 2. I don't believe this is explicitly mentioned in the text but should be. Is this the result of progressive branching?

Are PNS axons also smaller in diameter?

REVIEWERS' COMMENTS

Reviewer #1 (Remarks to the Author):

The authors have addressed my concerns and in my opinion the manuscript is ready for publication. This is a rigorous and exciting study.

Reviewer #1 (Remarks on code availability):

Acceptable

We thank the reviewer and are glad we have addressed all their comments.

Reviewer #2 (Remarks to the Author):

The authors have conscientiously addressed the concerns raised in the prior review, by performing additional quantification of the structural features of Importin13 deficient neurons and of myelin thickness. They have also expanded the discussion to speculate about the possible mechanisms responsible for the change in axon diameter in the mutants. These changes have significantly improved the study. My only lingering concern relates to their firm conclusion that axon diameter is the primary driver of the changes in conduction velocity observed. Since they do not provide mechanistic insight into the pathways regulated by Importin13 and this protein has known pleiotropic effects, this seems a likely oversimplification. That is, from the data provided it does not seem convincing that the observed effects can be attributed solely to changes in axon diameter as a physical property - related only to changes in capacitance and internal resistance. It seems reasonable that there may be many co-regulated genes that control conduction velocity (eg. ion channel density and composition, internode length, etc), and possible complex interactions between ion channel expression and axon diameter growth. Without manipulation of other possibilities, I think that this conclusion should be softened. From the response, I think that this could mostly be a semantic issue, but nevertheless one that could be confusing to readers. I suggest that the authors take a more neutral stance in the title and abstract about the mechanisms responsible for the conduction velocity change, perhaps by using the phrase "axon diameter and co-regulated processes" or something similar. Of note, I think the evidence that myelination does not appear to adapt to this reduction in diameter is fascinating and should be further highlighted in the abstract, as this runs counter to the prevailing evidence of axon diameter being a primary driver of myelin thickness.

We thank the reviewer for their support and also their considered view on how to ensure that our key conclusions are phrased appropriately. We agree with everything that the reviewer has noted, and had gone to some lengths to convey precisely these points in our revised discussion, noting that axon diameter may indeed be co-regulated with certain other features of neuronal

development that influence function. Similarly, we were very careful with our phrasing of our revised abstract, which concludes by saying “This suggests that axon diameter growth is a principal driver of increases in conduction speeds along myelinated axons over time;” we believe that stating “a principal driver” is much more measured than “the principal driver,” which we agree would represent a significant over-statement. In response to the reviewer’s suggestion, we did try to rework the title, but found that adding “...and other co-regulated processes” added confusion. In our opinion the title is already quite conservative, given that it is well established that axon diameter affects conduction. The principal novelty of the title notes the discovery that importin 13 influences axon diameter, and then, in turn conduction velocity. We hope that this is clear and conveys the topic and conclusions of the study sufficiently.

Reviewer #2 (Remarks on code availability):

This link was not associated with an active page on Github, so I was unable to review the code.

We thank the reviewer for bringing to our attention that the link was not working. This has been fixed.

Reviewer #3 (Remarks to the Author):

The authors have adequately considered my comments. In my view the manuscript is fit for publication.

We thank the reviewer and are glad we have addressed all their comments.

Reviewer #4 (Remarks to the Author):

This is a resubmission of a study that compellingly demonstrates importin 13 is required for radial expansion of large CNS axons in zebrafish while sparing the soma and dendrites. They further show the increase in axonal diameter is the major driver of increased conduction velocity independent of myelin thickness. The authors have been responsive to concerns raised in previous reviews including providing a rescue experiment for the mutant, characterizing myelination more completely, and revising the text/discussion. These changes have strengthened the manuscript. While the mechanisms by which importin 13 regulates axon diameter are unclear, the authors reasonably argue elucidation is beyond the scope of the current study.

I have one modest, lingering concern and otherwise minor issues the authors should address:

The evidence that conduction is dependent on axon diameter and not myelin thickness is mostly

convincing. However, other parameters that were not examined - including node width, internode length, and paranode integrity have all been implicated in regulating conduction velocity as the authors discuss. In particular, as axon diameter has been shown to be a determinant of internode length (Bechel et al, 2015) - which in turn impacts CV - this parameter seems particularly relevant to measure in their mutant, perhaps even from existing images. This data would also shed physiological insights into whether myelin thickness is indeed regulated by axon diameter in contrast to myelin thickness.

We agree with the reviewer that analyses of myelin sheath length in our mutants could be an interesting addition to the paper. However, we cannot obtain this data from existing images, in which all myelin is labelled, as we do not have the ability to visualize individual myelin sheaths. To obtain this data would require mosaic labelling of individual oligodendrocytes and identifying isolated sheaths on the Mauthner axon and measuring their length. We put a lot of effort into this exact experiment during the initial revision period, but unfortunately the very low throughput and high variability made it impossible to complete in a reasonable time frame. Trying to extend this analysis alone now would likely add months of work, and would not affect any of our conclusions, irrespective of the outcome.

However, we absolutely agree with the reviewer that as a field we need to move towards a much more holistic view of the myelinated axon, and that we need to deconstruct the relative influence of the various factors that influence distinct aspects of conduction (including myelin sheath/internode length and node widths as the reviewer notes). In our previous revision, we added significantly to our discussion to address this point. Working towards understanding the mechanisms that regulate the many conduction-regulating features of the myelinated axon, such as growth in diameter, the expression, delivery and localisation of ion channels and transporters to correct axonal domains, myelination, and more will require a huge amount of targeted investigation by the community in the future.

The authors show a graph quantifying rescue of axon diameter in Fig. 2J; it would be useful to show a micrograph of an example of the rescued axon as well in the figure

We have added representative images of the rescue experiment to figure 2 as requested by the reviewer.

Do they know if the original C-terminal ue57 mutant affects importin 13 function or does it primarily affect its abundance?

Unfortunately, we do not have an answer to this question. Antibodies targeting importin13 do not work in zebrafish, so we were not able to assess how much importin13b protein is present in these mutants.

The authors demonstrate there is no change in myelin thickness in the neuron-specific knockout. I may have missed it but is myelin thickness also unchanged in the original pan-mutant?

We chose to focus our analyses of myelination on neuron-specific mutants, as these are the mutants we use to assess conduction velocity. Furthermore, assessing myelin thickness in the full mutants would be confounded by the fact that importin 13 may play additional roles in oligodendrocytes; thus, changes to myelin in full mutants may or may not be due to changes in axon diameter, which is the focus of our paper. We are currently completing a separate study which assesses the role of importin 13 in myelinating cells, which will be published separately.

The diameter of the Mauthner axon tapers from proximal to distal based on examination of Supplemental Figure 2. I don't believe this is explicitly mentioned in the text but should be. Is this the result of progressive branching?

Yes, the diameter of the Mauthner axon does taper along the proximal to distal length of the axon, and we have added this to our discussion. It is not uncommon for axons to change diameter along their length (we speak to this in the introduction), and this has been observed for other types of neurons. Our analyses was not affected by this tapering because, as shown in Figure 1D and Supplemental Figure 2, and stated in the material and methods, we always performed our comparisons between controls and mutants at the same region along the length of the axon. The mechanisms responsible for differential axon diameter along the length of axons remain unknown – progressive branching is one possibility, but to our knowledge this idea has not been tested experimentally.

Are PNS axons also smaller in diameter?

As noted in our previous round of revisions, we do see that axon diameter is affected in the peripheral nervous system in the full ipo13b mutants. However, these mutants also have a severe reduction in the number of Schwann cells, meaning many axons that are normally myelinated are either unmyelinated or partially myelinated. Since we know Schwann cells are regulators of axon diameter in the PNS, it is difficult to disentangle whether the reduction in diameter is due to disruption of ipo13b or disruption to myelin in full mutants. We have not looked at the PNS in our neuron-specific ipo13b mutants; however, we expect this would be challenging due to the mosaicism in the CRISPR/Cas9 based model used in this study. This could be something we address in the future, by using recently published improved techniques in zebrafish that can disrupt gene function in a cell-type and temporal specific manner without the mosaicism inherent to the CRISPR/Cas9 based cell-type specific strategy used in this study. Interestingly, and for the reviewer's insight, we are currently in the process of putting together a manuscript which shows that oligodendrocytes/myelin in the CNS does not affect axon diameter growth (both in zebrafish and mice). This divergence in the mechanisms regulating

axon diameter in the CNS and PNS is one of the reasons we have chosen to only focus on the CNS in our paper.